# Heterozygote advantage can explain the extraordinary diversity of immune genes

Mattias Siljestam*, Claus Rueffler

Department of Ecology and Genetics, Animal Ecology, Uppsala University, Uppsala, Sweden

**Abstract** The majority of highly polymorphic genes are related to immune functions and with over 100 alleles within a population, genes of the major histocompatibility complex (MHC) are the most polymorphic loci in vertebrates. How such extraordinary polymorphism arose and is maintained is controversial. One possibility is heterozygote advantage (HA), which can in principle maintain any number of alleles, but biologically explicit models based on this mechanism have so far failed to reliably predict the coexistence of significantly more than 10 alleles. We here present an eco-evolutionary model showing that evolution can result in the emergence and maintenance of more than 100 alleles under HA if the following two assumptions are fulfilled: first, pathogens are lethal in the absence of an appropriate immune defence; second, the effect of pathogens depends on host condition, with hosts in poorer condition being affected more strongly. Thus, our results show that HA can be a more potent force in explaining the extraordinary polymorphism found at MHC loci than currently recognised.

## Editor's evaluation

This important theoretical and numerical study deals with a contemporary topic in evolutionary biology, immunology and population genetics. The structure of the models and the analytic framework used are relevant and sound, and the combination of two types of models is a powerful approach that produces compelling evidence to support the hypothesis on the role of heterozygote advantage in maintaining MHC gene polymorphism. The description of the models is easy to follow, and the paper would be of interest to specialists in evolution, immunology, and the general *eLife* readership.

*For correspondence:
m@siljestam.com

Competing interest: The authors declare that no competing interests exist.

## Introduction

Heterozygote advantage (HA) is a well-established explanation for single locus polymorphism, with the sickle cell locus as a classical textbook example (*Allison, 1954*). However, whether HA is generally important for the maintenance of genetic polymorphism is questioned (*Hedrick, 2012*; *Sellis et al., 2016*). Genes of the major histocompatibility complex (MHC), responsible for inducing immune defences by recognising the agretopes of the pathogenic antigens, are the most polymorphic loci among vertebrates (*Duncan et al., 1979*; *Apanius et al., 1997*; *Penn, 2002*; *Sommer, 2005*; *Eizaguirre and Lenz, 2010*). HA as an explanation for this high level of polymorphism was introduced almost 50 years ago by *Doherty and Zinkernagel, 1975*. The idea suggests that individuals with MHC-molecules from two different alleles are capable of recognising a broader spectrum of pathogens, resulting in higher fitness. This is especially evident when the MHC-molecules of the two alleles have complementary immune profiles (*Pierini and Lenz, 2018*), a phenomenon known as divergent allele advantage (*Wakeland et al., 1990*), and *Stefan et al., 2019* show that this allows for the coexistence of alleles with larger variation in their immune efficiencies. Early theoretical work suggested

that HA can maintain an arbitrarily high number of alleles if these alleles have appropriately fine-tuned homo- and heterozygote fitness values (*Kimura and Crow, 1964*; *Wright, 1966*; *Maruyama and Nei, 1981*). However, later work suggests that such genotypic fitness values are unlikely to emerge through random mutations (*Lewontin et al., 1978*). More mechanistic models have also failed to reliably predict very high allele numbers (*Spencer and Marks, 1988*; *Hedrick, 2002*; *De Boer et al., 2004*; *Borghans et al., 2004*; *Stoffels and Spencer, 2008*; *Trotter and Spencer, 2008*; *Trotter and Spencer, 2013*; *Ejsmond and Radwan, 2015*; *Lau et al., 2015*). As a result, HA plays only a minor role in current explanations of polymorphism at MHC loci (*Hedrick, 1998*; *Gould et al., 2004*; *Wegner, 2008*; *Kekäläinen et al., 2009*; *Eizaguirre and Lenz, 2010*; *Lenz, 2011*; *Loiseau et al., 2011*), despite empirical evidence for its existence (*Doherty and Zinkernagel, 1975*; *Hughes and Nei, 1989*; *Jeffery and Bangham, 2000*; *Penn et al., 2002*; *McClelland et al., 2003*; *Froeschke and Sommer, 2005*; *Kekäläinen et al., 2009*; *Oliver et al., 2009*; *Lenz, 2011*). Consequently, other mechanisms are suggested to be important for the maintenance of allelic diversity, such as Red-Queen dynamics, fluctuating selection, and disassortative mating (*Apanius et al., 1997*; *Hedrick, 1998*; *Penn, 2002*; *Borghans et al., 2004*; *Wegner, 2008*; *Spurgin and Richardson, 2010*; *Loiseau et al., 2011*; *Ejsmond and Radwan, 2015*; *Ejsmond et al., 2023*).

Our study challenges this status quo by demonstrating that HA is a potent force that can drive the evolution and subsequent maintenance of more than 100 alleles. To demonstrate that it is indeed HA that is responsible for allelic diversity in our model, we deliberately keep all aspects of the pathogen community fixed to exclude any Red-Queen dynamics. The novelty of our approach lies in the fact that we do not rely on hand-picked genotypic fitness values. Instead, these fitness values emerge from our eco-evolutionary models, where the allelic values that allow for extraordinary polymorphism are found by evolution in a self-organised process. We do not claim that HA is the only mechanism responsible for the diversity of MHC-alleles in nature. However, our results show that HA can be more important than currently recognised.

## Model

We investigate the evolution at an MHC locus using mathematical modelling and computer simulations. In the following sections, we describe how genotypes map to immune response and ultimately to survival, followed by a description of our evolutionary algorithm.

We assume that the MHC-molecules produced by the two alleles at a diploid MHC locus determine the immune response based on antigen recognition against multiple pathogens present in the environment. Our approach is based on the following two key assumptions regarding the relationship between pathogen virulence and host fitness:

a. Virulent pathogens are lethal in the absence of an appropriate immune defence.
b. The effect of pathogens on host survival depends on host condition, with hosts in poorer condition being affected more strongly.

An implication of the second assumption is that the combined effect of multiple pathogens on host survival exceeds the sum of the effects of each pathogen alone.

To incorporate these two assumptions, we assume that the effect of pathogen attacks on host survival acts through the intermediary step of the host's 'condition', which is a proxy for a suit of measurements describing an individual's body composition and physiology (*Wilder et al., 2016*). In the absence of an adequate immune response, a pathogen attack reduces the condition of a host to zero, causing its death (assumption a). More generally, we assume that the probability to survive is an increasing function of condition and that a host clearing a pathogen is in a weaker condition afterward. Since the survival probability cannot exceed one, the function that maps condition to survival has to be saturating. Consequently, for high values of conditions, where the survival function has saturated, pathogens reducing condition have small effects on survival. As condition decreases, pathogen-induced reductions have larger effects on survival (assumption b). A natural biological intuition for assumption (b) can be drawn from examples like COVID-19 or influenza, where it is well known that these pathogens do not pose a high mortality risk to individuals in good condition, but can significantly increase mortality risk for individuals in poor condition (*Thompson, 2004*; *Zhou et al., 2020*).

A further assumption of our model is the existence of a trade-off between the efficiencies of MHC-molecules to induce a defence against different pathogens. Thus, no MHC-molecule can perform

optimally with respect to all pathogens and an improved efficiency against one set of pathogens can only be achieved at the expense of a decreased efficiency against another set of pathogens. Under such trade-offs, an MHC-molecule can be specialised to detect a few pathogens with high efficiency, or, alternatively, be a generalist molecule that can detect many pathogens but with low efficiency. There is empirical support for the existence of such trade-offs. First, many MHC-molecules can detect only a certain set of antigens (*Wakeland et al., 1990*; *Froeschke and Sommer, 2012*; *Eizaguirre et al., 2012*; *Chappell et al., 2015*; *Pierini and Lenz, 2018*) and therefore provide different degrees of protection against different pathogens (*Wakeland et al., 1990*; *Apanius et al., 1997*; *Eizaguirre and Lenz, 2010*; *Froeschke and Sommer, 2012*; *Eizaguirre et al., 2012*; *Cortazar Chinarro et al., 2022*). Second, it has also been found that specialist MHC-molecules are expressed at higher levels at the cell surface while generalist MHC-molecules that bind less selectively are expressed at lower levels (*Chappell et al., 2015*), potentially to reduce the harm of binding self-peptides. This could explain the lower efficiency of generalist MHC-molecules.

We employ two approaches to model this trade-off. First, we use unimodal functions to model the match between MHC-molecules and pathogens. This approach has a long history in evolutionary ecology (e.g. *Levins, 1968*; *Sheftel et al., 2018*), and, when using Gaussian functions, the model becomes amenable to mathematical analysis. We envisage that these pathogen optima represent distinct pathogen species from diverse taxonomic groups such as fungi, viruses, bacteria, protists, helminths, and prions, among others (*Schmid-Hempel, 2021*). Hence, we expect these pathogen optima to remain approximately constant over the time scales considered in our model. By keeping all aspects of the pathogen community fixed, we exclude Red-Queen dynamics and ensure that the observed allelic polymorphism is driven solely by HA.

To demonstrate that the allelic diversity evolving in the Gaussian model does not dependent on the specificities of this model but rather results from the model fulfilling the above assumptions (a) and (b), we implement an alternative and more mechanistic approach to model pathogen recognition. Inspired by *Borghans et al., 2004*, in this approach, while keeping assumptions (a) and (b) intact, immune defence is based on the match between two binary strings (or bit-strings), one representing the MHC-molecule and the other a peptide of the pathogen. In this model, a single MHC-allele has the potential to detect several pathogens, which could be interpreted as the different pathogens being more closely related.

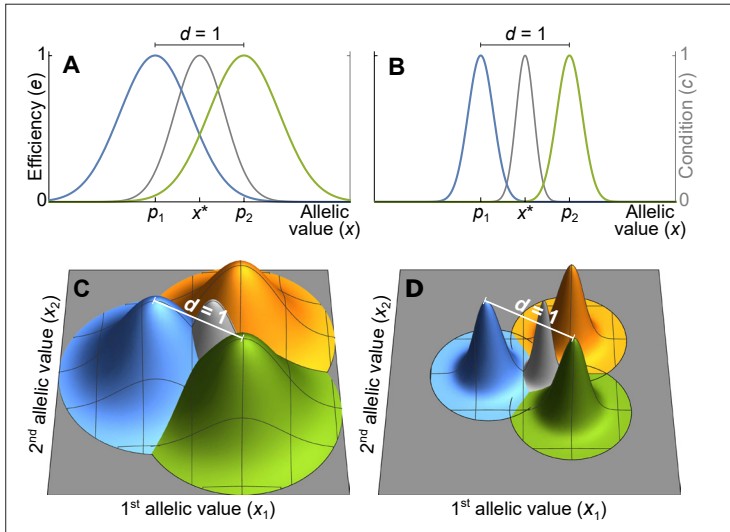

**Figure 1.** Efficiency against two pathogens (coloured lines in **A–B**) and three pathogens (coloured cones in **C–D**) as a function of allelic values $x$. Efficiencies are modeled with Gaussian functions with pathogen optima at equal distances $d = 1$ (indicated by $p_1$ and $p_2$ in **A, B**). The width of the Gaussian functions, which determine how severely pathogens affect hosts with suboptimal major histocompatibility complex (MHC) molecules, is given by the virulence parameter $v$. With high virulence ($v = 7$, narrow Gaussians in **B, D**), alleles away from the optima have a low efficiency, while for a low virulence ($v = 2.5$, wide Gaussians in **A, C**) efficiency is higher. Grey lines and cones give the condition $c$ of homozygote individuals. The generalist allele, maximising condition, is located at the centre with equal distance to all pathogen optima (indicated by $x^*$ in **A, B**).

By explicitly modelling MHC efficiencies against various pathogens – rather than assuming a fixed proportion of pathogens detected per MHC-molecule (as, e.g., *De Boer et al., 2004*) – our model accounts for the possibility that MHC-molecules can have complementary immune profiles. When paired, complementary alleles produce fit heterozygotes able to detect an increased number of pathogen peptides (*Pierini and Lenz, 2018*), exemplifying the concept of divergent allele advantage in the sense of *Wakeland et al., 1990*.

## Gaussian model

In this approach, we use Gaussian functions to model the ability of MHC-molecules to recognise $m$ different pathogens, as illustrated in *Figure 1*. Here, MHC-alleles and pathogens are represented by vectors $x = (x_1, x_2, \ldots, x_h)$ and $p_k = (p_{1k}, p_{2k}, \ldots, p_{hk})$, respectively. The MHC-alleles code for MHC-molecules, and the ability of an MHC-molecule to recognise the $k$th pathogen is maximal if $x = p_k$. This ability decreases with increasing distance between $x$ and $p_k$. The decrease is modelled using an $h$-dimensional Gaussian function, as detailed in *Equation A14*. The nature of the trade-off can be varied by adjusting the positions of the pathogen optima and the shape of the Gaussian functions.

Without loss of generality, we can reduce the dimension of the vectors $x$ and $p_k$ to $h = m - 1$, such that $x = (x_1, x_2, \ldots, x_{m-1})$ and $p_k = (p_{1k}, p_{2k}, \ldots, p_{m-1,k})$. For example, in *Figure 1A and B*, where $m = 2$, the x-axis represents the unique line passing through two pathogen optima in a trait space of potentially much higher dimension. Similarly, in *Figure 1C and D*, where $m = 3$, the two-dimensional coordinate system represented by the grey surfaces describes the unique plane passing through three pathogen optima. Mathematically speaking, $m$ linearly independent pathogen optima form the basis of a vector space of dimension $m - 1$, which we choose as the coordinate system for the vectors $x$ and $p$. Allelic vectors outside this set are necessarily maladapted for all pathogens along at least one dimension, and owing to our dimensionality reduction we ignore such trait vectors.

We examine two versions of the Gaussian model. The first one is based on two symmetry assumptions and shown in *Figure 1*: pathogen optima are placed symmetrically such that the distance between any two pathogens equals 1, and the Gaussian functions $e_k(x)$ are isotropic (rotationally symmetric) and of equal width. This allows to simplify the covariance matrix in the Gaussian function $e_k(x)$ (*Equation A14*) such that it can be replaced with a single parameter $v$ (Appendix 'Model description'),

$$e_k(x) = \exp\left(-\frac{v^2}{2}(x - p_k)^{\mathrm{T}}(x - p_k)\right),$$ (1)

where the superscript T indicates vector transposition. The parameter $v$, to which we refer as virulence, is the inverse of the width of the Gaussian function. If the Gaussian function is narrow, corresponding

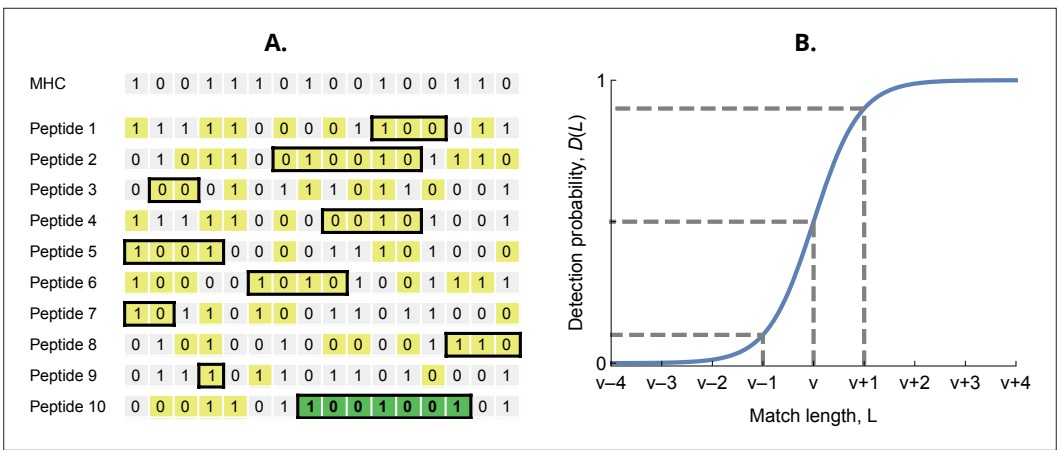

**Figure 2.** Detection probability in the bit-string model. (**A**) Major histocompatibility complex (MHC) bit-string matching against a pathogen with $n_{\mathrm{pep}} = 10$ peptides. Yellow indicates a match between MHC and peptide bits. The longest consecutive match per peptide ($L$) is indicated with a black box. The longest match over all peptides occurs for the last peptide, marked in green, with match length $L = 7$. (**B**) Detection probability for peptides as a function of match length $L$ (*Equation 2* with $a = \log(9)$). The dashed lines indicate, from left to right, 10%, 50%, and 90% detection probability.

to a high virulence $v$, a pathogen causes significant harm if MHC-molecules are not well adapted against it (*Figure 1B and D*). On the other hand, if the Gaussian function is wide, corresponding to a low virulence $v$, a pathogen causes less harm (*Figure 1A and C*).

We relax these symmetry assumptions in the second version, where we allow for Gaussian functions with arbitrary shape and position. Since the results for the two versions are similar, we here focus on the case with symmetry and refer to Appendices 'Deviations from symmetry in the Gaussian model', 'Mode description', and 'Analytical results for the Gaussian model' for results based on general Gaussian functions.

## Bit-string model

Our second approach is inspired by *Borghans et al., 2004*, and commonly referred to as a bit-string model. Pathogens are assumed to produce $n_{pep}$ peptides, and for a pathogen to cause virulence, all of its peptides have to avoid detection by the host's MHC-molecules. We here equate MHC-alleles with the MHC-molecule they code for, and both MHC-molecules and pathogen peptides are represented by binary strings (or bit-strings) of, following *Borghans et al., 2004*, length 16.

The probability that an MHC-molecule detects a pathogen peptide increases with the maximum match length of consecutive matches between their binary strings. For an MHC-molecule $\boldsymbol{x}$ and the $k$th peptide of the $i$th pathogen, this match length is denoted $L_{ki}(\boldsymbol{x})$, or $L$ for short (see *Figure 2A*). The corresponding detection probability, denoted $D(L_{ki}(\boldsymbol{x}))$, is then given by the logistic function

$$D(L_{ki}(\boldsymbol{x})) = \frac{1}{1 + \exp[a(v - L_{ki}(\boldsymbol{x}))]}. \tag{2}$$

Here, $v$ denotes the required match length $L$ for a 50% chance of detection. The parameter $v$ has again the interpretation of virulence, with higher values indicating pathogen peptides that are harder to detect by MHC-molecules. The positive parameter $a$ governs the steepness of the function $D$. We choose $a = \log(9)$, which results in $D(L)$ equalling 10% when $L = v - 1$ and 90% when $L = v + 1$ (*Figure 2B*). Finally, the realised efficiency of an MHC-molecule $\boldsymbol{x}$ against the $k$th pathogen is given by the probability of detecting at least one of its $n_{pep}$ peptides, which equals

$$e_k(\boldsymbol{x}) = 1 - \prod_{i=1}^{n_{pep}} \left(1 - D(L_{ki}(\boldsymbol{x}))\right). \tag{3}$$

## From immune defence to survival

For both versions of our model, we assume that MHC-alleles are co-dominantly expressed (*Eizaguirre and Lenz, 2010*; *Abbas et al., 2014*), and an individual's efficiency to recognise pathogens of type $k$ is given by the arithmetic mean of the efficiencies from its two alleles. We want to note that assuming co-dominance gives more conservative results in terms of the number of coexisting alleles, as dominance would increase the degree of HA.

For each pathogen attack, an individual's condition $c$ is reduced by a certain fraction that depends on the efficiency of the defence $e$ against that pathogen. Since each individual is exposed to all pathogens during their lifetime, the condition $c$ is determined by the product of its defences against all pathogens,

$$c(\boldsymbol{x}_i, \boldsymbol{x}_j) = c_{max} \prod_{k=1}^{m} \frac{e_k(\boldsymbol{x}_i) + e_k(\boldsymbol{x}_j)}{2}, \tag{4}$$

where $\boldsymbol{x}_i$ and $\boldsymbol{x}_j$ represent the MHC-alleles the host carries at the focal locus, and $c_{max}$ is the condition of a hypothetical individual with perfect defence against all pathogens (see Appendix 'Model description' for more details). Because $e_k(\boldsymbol{x}) < 1$, condition is reduced with each additional pathogen in a proportional manner. The multiplicative nature of *Equation 4* has the effect that a poor defence against a single pathogen is sufficient to severely compromise condition, and therefore survival (see next paragraph), fulfilling assumption (a) above.

Finally, survival $s$ is an increasing but saturating function of an individual's condition $c$,

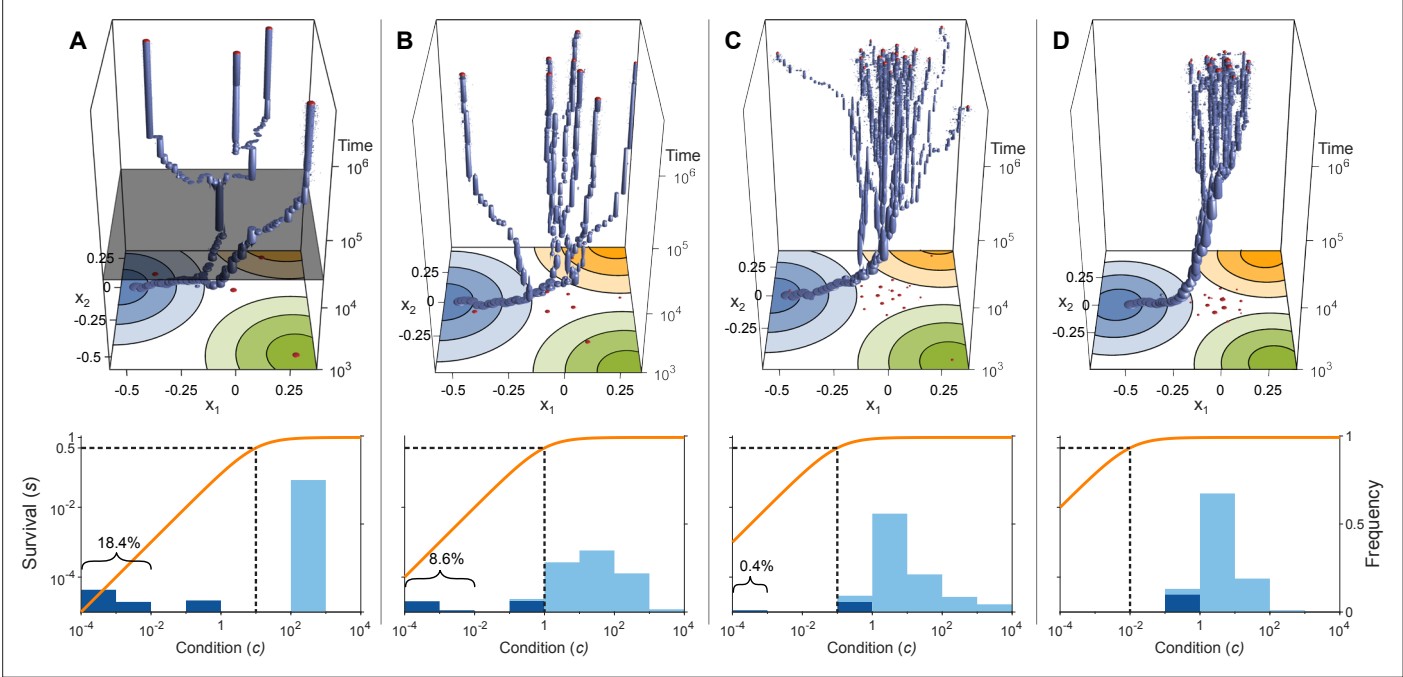

**Figure 3.** Evolution of allelic values under the Gaussian model in the presence of three pathogens (arranged as in **Figure 1D**) for four different values of the survival half-saturation constant $K$ (**A**: $K = 10$, **B**: $K = 1$, **C**: $K = 0.1$, **D**: $K = 0.01$; dashed line in lower panel). The top panel shows individual-based simulations. The two horizontal axes give the two allelic values $x = (x_1, x_2)$ that characterise an allele, while the vertical axis shows evolutionary time. The thickness of the blue tubes is proportional to allele frequencies. Allelic values at the last generation are projected as red dots on the top as well as on the bottom plane. Coloured circles represent the contour lines of the Gaussian efficiency functions $e_k(x)$ as shown in **Figure 1D**. In all simulations, gradual evolution leads towards the generalist allele $x^* = (0, 0)$ and branching occurs in its neighbourhood, as predicted by our analytical derivations (Appendix 'The evolutionarily singular point'). In (**A**) there are three consecutive branching events with the second branching event marked by the grey plane ($n_e = 4.0$; for details regarding $n_e$, see the legend of **Figure 4**). (**B and C**) show that, as $K$ decreases, the number of branching events increases, resulting in more coexisting alleles ($n_e = 7.8$ and $n_e = 16.5$, respectively). Finally, (**D**) reveals that, as $K$ decreases even further such that already low condition values result in high survival, the number of branching events decreases again, resulting in a set of alleles closely clustered around the generalist allele ($n_e = 10.2$). The bottom panel shows survival $s$ as a function of condition $c$ as defined by **Equation 5** on a log-log scale (orange line, left vertical axis) and the frequencies of individual conditions at the final generation (dark blue bars for homozygotes and light blue bars for heterozygotes, right vertical axis; conditions from 0 to $10^{-4}$ are incorporated into the first bar). These panels show that increased allelic diversity results in a lower proportion of homozygote individuals, which have lower survival. Other parameter values: $v = 7$, $N = 2 \times 10^5$, $\mu = 10^{-6}$, and $\delta = 0.016$.

$$s(x_i, x_j) = \frac{c(x_i, x_j)}{K + c(x_i, x_j)}. \tag{5}$$

Here, $K$ is the survival half-saturation constant, giving the condition $c$ required for a 50% chance of survival. This function fulfils assumption (b) above as long as $K$ is not too large. Individuals in good health then have a condition $c$ far above $K$, and a decrease in condition only has a small effect on survival. If $c$ is lower than $K$, then the host is in bad health and any additional pathogen causes a large reduction in survival $s$ (orange lines in bottom panel of **Figure 3** and 6).

In summary, **Equations 4 and 5** entail that assumptions (a) and (b), as formulated above, are satisfied. Using two distinct models to describe the interaction of hosts and pathogens, which both impose a trade-off between the ability to detect different pathogens – namely the Gaussian and the bit-string model – we demonstrate below that HA emerges as a potent force capable of driving the evolution of a very high number of coexisting alleles.

## Analysis

To study the evolutionary dynamics of allelic values $x$ in both the Gaussian and the bit-string model, we simulate a diploid Wright-Fisher model with mutation and selection (**Fisher, 1930**; **Wright, 1931**). Thus, we consider a diploid population of fixed size $N$ with non-overlapping generations and random mating. Individuals produce, independent of their genotype, a large number of offspring, resulting in

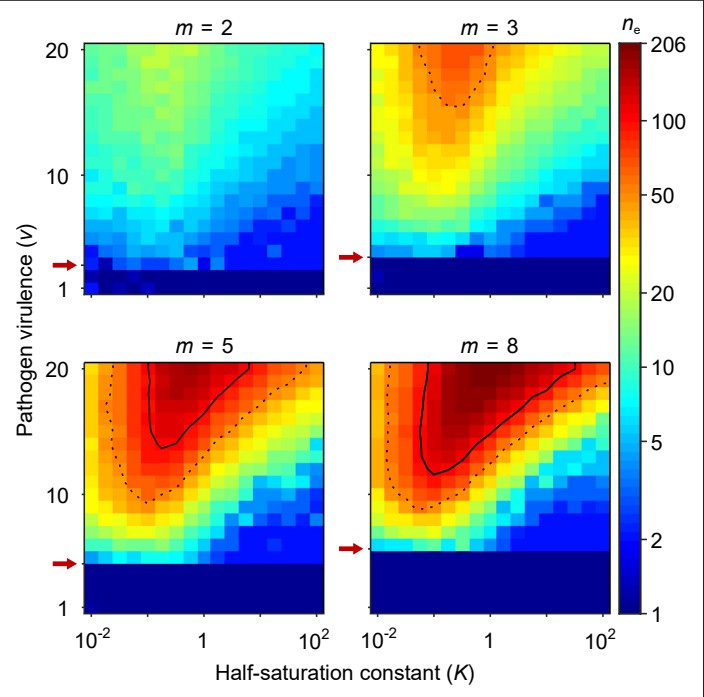

**Figure 4.** Number of coexisting alleles under the Gaussian model for $m$ pathogens as a function of pathogen virulence $v$ and the survival half-saturation constant $K$. Figures are based on a single individual-based simulation per pixel and run for $10^6$ generations, assuring that the equilibrium distribution of alleles is reached. Results are reported in terms of the effective number of alleles $n_e$, which is a conservative measure for the number of alleles, discounting for rare alleles present at mutation-drift balance (see Appendix 'Effective number of alleles'). The clear pattern in the figures indicates a high degree of determinism in the simulations. Using population size $N = 10^5$ and per capita mutation probability $\mu = 5 \times 10^{-7}$, the expected $n_e$ under mutation-drift balance alone equals 1.2 (see Appendix 'Effective number of alleles'). Dashed and solid lines give the contours for $n_e = 50$ and $n_e = 100$, respectively. Red arrows indicate $v = 2\sqrt{m}$, the threshold for polymorphism to emerge from branching (**Equation A46**). Accordingly, simulations in the dark blue area result in a single abundant allele with $n_e$ close to one. Other parameters: expected mutational step size $\delta = 0.03$.

deterministic Hardy-Weinberg proportions before viability selection. After viability selection, which is based on **Equation 5** and adjusts the proportion of genotypes accordingly, stochasticity is introduced by random multinomial sampling of $N$ surviving offspring, which constitute the adult population of the next generation. Using this model, we follow the fate of recurrent mutations that occur with a per capita mutation probability $\mu$. The long-term evolutionary dynamics is obtained by iterating this procedure (**Figure 3**, top panel) until the number of alleles equilibrates. This procedure can result in high numbers of coexisting alleles, where the emerging allelic polymorphism is driven by increasing the alleles' expected survival (or marginal fitness, see **Equations A5–A6** in Appendix 'Adaptive dynamics and invasion fitness').

For the Gaussian model, mutations are drawn from an isotropic normal distribution with expected mutational effect size $\delta$ (Appendix 'Varying the expected mutational step size in the Gaussian model'). We here focus on mutations of small effect ($\delta = 0.016$ in **Figure 3** and $\delta = 0.03$ in **Figure 4**) and thus near-gradual evolution. The effect of smaller and larger effect sizes is investigated in Appendix 'Varying the expected mutational step size in the Gaussian model'. To minimise computation time, simulations (other than those in **Figure 3**) are initialised at the trait vector that is given by the mean of the vectors describing the pathogens. In the bit-string model, the $m$ pathogens are each given $n_{pep}$ randomly drawn bit-strings at the beginning of a simulation and the host population is initialised with a single MHC-allele given by a randomly drawn bit-string. Mutations change a random bit of the MHC-allele. The bit-string model can indeed only be analysed with computer simulations. In contrast, for the Gaussian model we can analytically derive conditions under which to expect either a single generalist allele or the build-up of allelic diversity through gradual evolution in a process known as

evolutionary branching (*Metz et al., 1992*; *Geritz et al., 1998*; *Kisdi and Geritz, 1999*; *Doebeli, 2011*) (see Appendices 'Mathematical analysis of the Gaussian model: Preliminaries' and 'Analytical results for the Gaussian model' for details).

## Results

### Gaussian model

In the simulations of the Gaussian model, the evolutionary dynamics first proceed towards a generalist allele with an intermediate efficiency against all pathogens, to which we refer to as $x^*$. This generalist allele maximises the condition $c$ for homozygote genotypes (grey lines and cones in *Figure 1*, Appendix 'Absolute convergence stability'). Once this generalist allele is reached, the evolutionary dynamics either stops (*Appendix 2—figure 1*), resulting in a population where all individuals are homozygous for $x^*$, or allelic diversification ensues (*Figure 3*), resulting in the coexistence of specialist and generalist alleles. Based on the adaptive dynamics approximation, we show analytically (Appendix 'The evolutionarily singular point') that $x^*$ is given by the arithmetic mean of the vectors $p_1, \ldots, p_m$ describing the $m$ pathogen optima (see *Equation A26* in Appendix 'The evolutionarily singular point') and an attractor of any sequence of allelic substitutions. Whether $x^*$ is an evolutionary stable endpoint or an evolutionary branching point where diversification ensues depends on the covariance matrix $\Sigma_p^2$ of the pathogen optima relative to the covariance matrices $\Sigma_G^2$ of the Gaussian efficiency functions (Appendices 'Derivation of the Hessian matrix of invasion fitness', 'Special case: identically shaped Gaussian efficiency functions', and 'Special case: maximal symmetry'). For the special case of identically shaped Gaussian functions, diversification occurs if and only if

$$\Sigma_p^2 - 2\Sigma_G^2 > 0, \tag{6}$$

(Appendix 'Special case: identically shaped Gaussian efficiency functions'). Note, that this expression is independent of the number of pathogens $m$. Under the additional assumption of equally distant pathogens and isotropic Gaussian functions, these covariance matrices are diagonal matrices with identical diagonal entries $\sigma_p$ and $\sigma_G$, respectively, and *Condition 6* simplifies to $\sigma_p^2 - 2\sigma_G^2 > 0$. When pathogen optima have an equal distance of 1, the variance among the optima $\sigma_p^2$ decreases with an increasing number of pathogens $m$, and the condition for evolutionary branching can be rewritten as

$$v > 2\sqrt{m}, \tag{7}$$

where $v = \sigma_G^{-1}$ (Appendix 'Special case: maximal symmetry').

*Figure 4* presents the final number of coexisting alleles as derived from individual-based simulations. It shows that the number of coexisting alleles increases with the number of pathogens $m$ and their virulence $v$, but also depends on the survival half-saturation constant $K$ (*Equation 5*). For a large part of the parameter space, more than 100 (solid contour lines in *Figure 4*) and up to over 200 alleles can emerge and coexist.

In order to better understand the process of allelic diversification, it is useful to inspect our analytical results in more detail. Evolutionary diversification occurs if mutant alleles $x'$ exist that can invade a population that is monomorphic for the generalist allele $x^*$. Initially, while still rare, such mutant alleles will always occur in heterozygous individuals, where they are paired with the generalist allele. Thus, our condition for evolutionary diversification, $v > 2\sqrt{m}$, is equivalent to $s(x', x^*) > s(x^*, x^*)$. Since, as homozygotes, the generalist allele maximises condition and therefore survival (Appendix 'Absolute convergence stability'), we also have $s(x^*, x^*) > s(x', x')$. In conclusion, individuals heterozygous for $x'$ and $x^*$ have higher survival than either homozygote, $s(x', x^*) > s(x^*, x^*) > s(x', x')$, and a polymorphism of these two alleles is maintained by HA, as suggested by *Doherty and Zinkernagel, 1975*. Furthermore, the generalist allele is an evolutionary branching point in the sense of adaptive dynamics theory (*Geritz et al., 1998*; *Kisdi and Geritz, 1999*).

The left-hand side of the diversification condition given by *Equation 7* indicates that invasion of more specialised alleles is favoured when pathogen virulence $v$ is large (narrow Gaussian functions, see *Figure 1A and C*). In this case, homozygotes for the generalist allele $x^*$ are relatively poorly protected against pathogens and more specialised alleles enjoy a fitness advantage while invading. The opposite is true when $v$ is small (wide Gaussian functions, see *Figure 1A and C*). The right-hand side of

the diversification criterion indicates that the benefit of specialisation decreases with an increasing number of pathogens (compare position of red arrows in *Figure 4*), because different pathogens require different adaptations. Thus, counter to intuition, initial allelic diversification is disfavoured in the presence of many pathogens.

If initial allelic diversification occurs, it leads to a dimorphism from which new mutant alleles can invade if they are more specialised than the allele from which they originated. Then, two allelic lineages emerge from the generalist allele $x^*$ and subsequently diverge (*Figure 3A*, up to $t = 3 \times 10^4$ below grey plane). Increasing the difference between the two alleles present in such a dimorphism has two opposing effects. The condition and thereby the survival of the heterozygote genotype increases because the MHC-molecules of the two more specialised alleles provide increasingly better protection against complementary sets of pathogens, i.e., these alleles are subject to a divergent allele advantage (*Wakeland et al., 1990*; *Pierini and Lenz, 2018*). On the other hand, survival of the two homozygote genotypes decreases because they become increasingly more vulnerable to the set of pathogens for which their MHC-molecules do not offer protection. Note that, due to random mating and assuming equal allele frequencies, half of the population are high survival heterozygotes and the remaining half homozygotes with low survival. Since survival is a saturating function of condition $c$ (*Equation 5*), it follows that the increase in survival of heterozygotes slows down with increasing condition (plateau of the orange curves in *Figure 3*), and the two opposing forces eventually balance each other such that divergence comes to a halt. At this point, our simulations show that the allelic lineages can branch again, resulting in three coexisting alleles. As a result, the proportion of low survival homozygotes decreases, assuming equal allele frequencies, from one-half to one-third. Subsequently, the coexisting alleles diverge further from each other because the increase in heterozygote survival once again outweighs the decreased survival of the (now less frequent) homozygotes (see *Figure 3A*, at time $t = 3 \times 10^4$, grey plane). In *Figure 3A*, this process of evolutionary branching and allelic divergence repeats itself one more time, resulting in four coexisting alleles. Consequently, 10 genotypes emerge: four homozygotes and six heterozygotes. The homozygotes with specialist alleles have a condition, and thereby a survival, close to zero (two left bars in bottom panel). Conversely, the homozygote for the generalist allele $x^*$ has an intermediate condition (middle bar), and all heterozygote genotypes have a survival close to 1 (right bar).

In *Figure 3B–D*, the process of evolutionary branching and allelic divergence continues to recur. As a consequence, allelic diversity continues to increase while simultaneously the proportion of vulnerable homozygote genotypes decreases (*Figure 3*, lower panel). Thus, in contrast to prior approaches (e.g. *Kimura and Crow, 1964*; *Wright, 1966*; *Lewontin et al., 1978*; *Maruyama and Nei, 1981*), we do not rely on hand-picked genotypic fitness values. Instead, in our approach, fitness values emerge from an eco-evolutionary model where evolution can be viewed as a self-organising process finding large sets of alleles that can coexist (*Figure 3*, upper panel).

We note that the half-saturation constant $K$ does not appear in the branching condition and thus does not affect whether polymorphism evolves. However, $K$ does affect the final number of alleles, which is maximal for intermediate values of $K$. This can be understood as follows. If $K$ is very large (right-hand side of the panels in *Figure 4*), then heterozygote survival saturates more slowly with increased allelic divergence so that continued allelic divergence is less counteracted. This hinders repeated branching (compare A and C in *Figure 3*). On the other hand, if $K$ is very small (left-hand side of the panels in *Figure 4*), then homozygous individuals can have high survival, which decreases the selective advantage of specialisation, leading to incomplete specialisation and a reduced number of branching events (compare D and C in *Figure 3*).

In summary, high virulence $v$ promotes allelic diversification. Increasing the number of pathogens $m$ has a dual effect: it hinders initial diversification but facilitates a higher number of coexisting alleles if diversification occurs, especially, for intermediate values of the half-saturation constant $K$.

We perform several robustness checks. First, *Appendix 4—figure 1* shows simulations in which we vary the expected mutational step size. These simulations show that the gradual build-up of diversity occurs most readily as long as the mutational step size is neither very small, since then the evolutionary dynamics becomes exceedingly slow, nor very large, since a large fraction of the mutants are then deleterious and end up outside the simplex made up of the pathogen optima (e.g. outside the triangle made up by the three pathogen optima in *Figure 1C and D*) so that they perform worse against all pathogens.

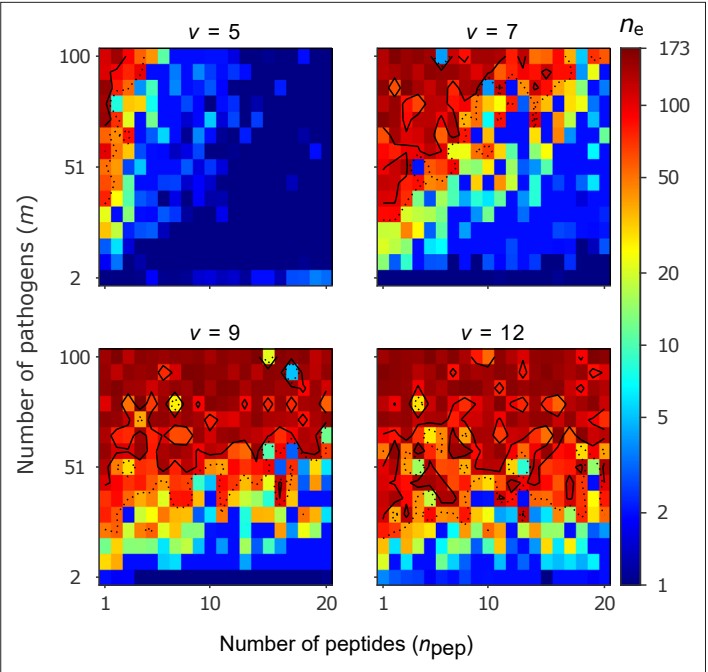

**Figure 5.** Number of coexisting alleles for the bit-string model for four values of virulence $v$ as a function of the number of pathogens $m$ (increased in steps of 7) and the number of peptides per pathogen $n_{pep}$. Figures are based on a single individual-based simulation per pixel and run for $10^6$ generations. Results are reported in terms of the effective number of alleles $n_e$, which discounts for rare alleles present at mutation-drift balance (see Appendix 'Effective number of alleles'). Using population size $N = 10^5$ and per capita mutation probability $\mu = 5 \times 10^{-6}$, the expected $n_e$ under mutation-drift balance alone equals 3. Dashed and solid lines give the contours for $n_e = 50$ and $n_e = 100$, respectively. Evolution started from populations monomorphic for a random allele, and run for $2 \times 10^6$ generations, assuring that the equilibrium distribution of alleles is reached. Other parameters: half-saturation constant $K = 1$.

Second, the results presented in *Figures 3 and 4* are based on the assumptions of equally spaced pathogen optima and equal width and isotropic Gaussian functions $e_k(\boldsymbol{x})$ as shown in *Figure 1*. In Appendices 'Analytical results for the Gaussian model' and 'Deviations from symmetry in the Gaussian model', we present analytical and simulation results, respectively, for the non-symmetric case. In particular, *Appendix 5—figure 1* shows that the predictions for the number of coexisting alleles presented here are qualitatively robust against deviations from symmetry. This is in line with *Condition 6* and its simplification under full symmetry, $\sigma_p^2 - 2\sigma_G^2 > 0$, showing that the more general condition for the evolution of allelic polymorphism is structurally identical to the condition under full symmetry.

### Bit-string model

Evolutionary diversification of MHC-alleles in the bit-string model is analysed with individual-based simulations, and the results are summarised in *Figure 5*. Similar to the Gaussian model, we find high levels of allelic polymorphism, with over 100 alleles coexisting in a significant portion of the parameter space. Note that we here keep the half-saturation constant $K$ fixed at 1. With this choice, the realised conditions occur both in the range where survival changes drastically with condition and where the survival function saturates (*Figure 6*), fulfilling assumption (b). This allows us to focus on the effect of the number of peptides $n_{pep}$ per pathogen.

Our results can be understood as follows. The likelihood that an MHC-molecule can recognise all pathogens is high in the following regions of the parameter space. Firstly, if virulence $v$ is low, then peptide recognition is more likely (*Equation 2*). Secondly, if the number of pathogens $m$ is low, then detection of all pathogens is a simpler task. Thirdly, if the number of peptides $n_{pep}$ per pathogen is high, then the potential for successful pathogen detection increases (*Equation 3*). Although our model is not sufficiently mechanistic to be directly related to parameters observed in nature, it suggests that when pathogens have a high number of peptides, maintaining allelic polymorphism requires a larger

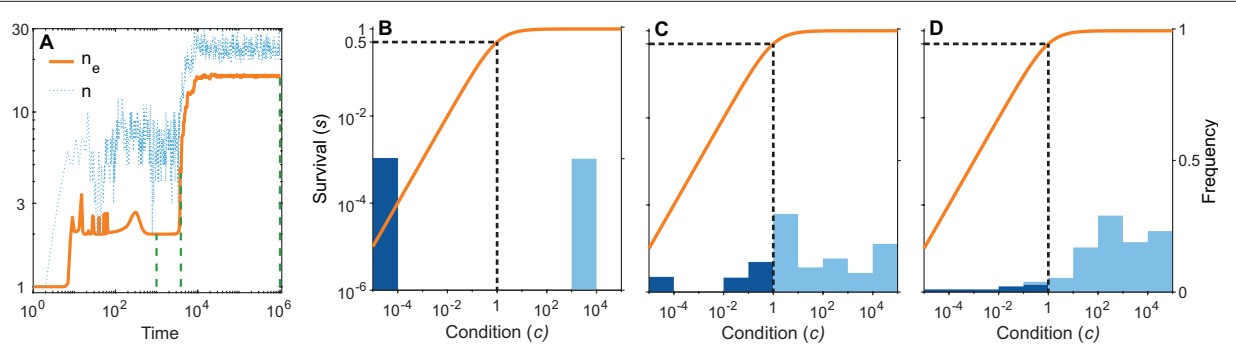

**Figure 6.** A simulation run showing the evolution of allelic diversity under the bit-string model in the presence of $m = 12$ pathogens. (**A**) shows the number of alleles $n$ and the effective number of alleles $n_e$ as a function of time (on a log-log scale). (**B–D**) give survival $s$ as a function of condition $c$ as defined by *Equation 5* on a log-log scale (orange line, left vertical axis) and the distribution of conditions at three time points (**B** $t = 100$, **C** $t = 3900$, **D** $t = 10^6$; vertical green dashed lines in **A**), with dark blue bars for homozygotes and light blue bars for heterozygotes (right vertical axis; conditions from 0 to $10^{-4}$ are incorporated into the first bar, and conditions from $10^4$ and greater are incorporated in the last bar). This shows that as allelic diversity increases, the frequency of homozygotes with low survival decreases. The black dashed lines indicate the value of $K = 1$. Other parameter values: $v = 7$, $m = 12$, $n_{pep} = 3$, $N = 10^5$, $\mu = 5 \times 10^{-6}$.

number of pathogens under conditions of low virulence ($v \leq 7$). For higher virulence ($v \geq 9$), the effect of $n_{pep}$ weakens, and allelic polymorphism evolves seemingly independent of the number of pathogens. Each of these three circumstances facilitates the existence of a single best allele whose MHC-molecule recognises all pathogens with a high probability (dark blue regions in *Figure 5*).

As virulence or the number of pathogens increases, or as the number of peptides decreases, the task of recognising all pathogens with high probability becomes progressively more challenging. This leaves homozygous individuals vulnerable to an increasing array of pathogens. As homozygotes get more vulnerable, there is a growing advantage for heterozygotes carrying alleles with complementary immune profiles, as these are able to detect up to twice as many pathogens as either homozygote. This increasingly stronger HA, in turn, facilitates coexistence of an increasing number of alleles, illustrated by increasingly warmer colours in *Figure 5*, and thereby decreases the proportion of vulnerable homozygotes. Thus, similar to the Gaussian model, increasing either the virulence $v$ or the number of pathogens $m$ enables a higher number of alleles to coexist. However, unlike the Gaussian model, increasing $m$ actually facilitates initial diversification rather than hindering it.

Importantly, in the bit-string model, a point mutation, switching the value of an arbitrary bit in the bit-string, can drastically alter the efficiencies against a large set of pathogens. Because of this, and in contrast to the Gaussian model, a polymorphism maintained by HA can emerge from many different alleles. On the other hand, gradual evolution in the Gaussian model is more efficient in finding the evolutionary endpoint of complementary alleles (*Figure 3*), while for the bit-string model, as the number of alleles increases, this becomes slower due to the lack of fine-tuning as mutations have large effect. To compensate for this, we use, compared to the Gaussian model, a higher mutation probability $\mu$ and run simulations for more generations.

*Figure 6A* shows the build-up of allelic diversity over time in an exemplary simulation run, and *Figure 6B–D* show the distribution of condition values at three time points, as indicated by green hatched lines in *Figure 6A*. In *Figure 6B* the population is dimorphic. Due to random mating, half of the population consists of homozygotes with low condition (dark blue bar), while the remaining half are heterozygotes with high condition (light blue bar). As time proceeds, the number of coexisting alleles increases. *Figure 6C* depicts a stage with five coexisting alleles (with at least 1% frequency) and an effective number of alleles ($n_e$) of 4.4. Ultimately, evolution results in 19 coexisting alleles (with at least 1% frequency), and an $n_e$ of 16.1, as shown in *Figure 6D*. In this process, the number of low condition homozygotes decreases, as indicated by the dark blue bars.

## Discussion

HA as an explanation for the coexistence of a large number of alleles at a single locus has a long history in evolutionary genetics. *Kimura and Crow, 1964*, and subsequently *Wright, 1966* showed that HA can in principle result in the coexistence of an arbitrary number of alleles at a single locus if two conditions are met: (1) all heterozygotes have a similarly high fitness, and (2) all homozygotes have a similarly low fitness. One special class of genes fulfilling these assumptions occur at self-incompatibility loci, where mating partners need to carry different alleles for fertilisation to be successful (*Wright, 1939*; *Castric and Vekemans, 2004*), or loci where homozygosity is lethal (*Ding et al., 2021*). However, more generally these conditions were deemed unrealistic by Kimura, Crow, and Wright themselves. This assessment was subsequently confirmed by *Lewontin et al., 1978*, who investigated a model in which the exact fitnesses are determined by drawing random numbers in a manner that all heterozygotes are more fit than all homozygotes. They found that the proportion of fitness arrays that leads to a stable equilibrium of more than six or seven alleles is vanishingly small. Similarly, the idea that the high allelic diversity found at MHC loci can be explained by HA was initially accepted by theoreticians (e.g. *Maruyama and Nei, 1981*; *Takahata and Nei, 1990*), but several later authors studying models based on more mechanistic assumptions were unable to reliably predict the coexistence of significantly more than 10 alleles (*Spencer and Marks, 1988*; *Hedrick, 2002*; *De Boer et al., 2004*; *Borghans et al., 2004*; *Stoffels and Spencer, 2008*; *Trotter and Spencer, 2008*; *Trotter and Spencer, 2013*; *Ejsmond and Radwan, 2015*; *Lau et al., 2015*). Thus, currently HA is largely dismissed as an explanation for highly polymorphic loci (*Gould et al., 2004*; *Eizaguirre and Lenz, 2010*; *Lenz, 2011*; *Hedrick, 2012*).

Our study, while not meant to be a highly realistic mechanistic representations of the interaction between MHC genes and pathogens, serves as a proof of principle that a high number of alleles, matching those found at MHC loci in natural populations, can indeed arise in an evolutionary process driven by HA. Our results thus revive the idea that HA has the potential to explain extraordinary allelic diversity. Importantly, and in contrast to several of the above-mentioned studies, this is achieved without making direct assumptions about homozygote and heterozygote fitnesses. Instead, our results emerge from two assumptions about how pathogens affect a host's condition and how this, in turn, affect survival. Assumption (a) states that pathogens are lethal in the absence of an appropriate immune response. This assumption is implemented in our model by assuming that each pathogen decreases a host's condition in a proportional manner (*Equation 4*), rather than by a fixed amount. Assumption (b) states that the effect of pathogens depends on host condition, with hosts in poorer condition being affected more strongly. Then, the combined effect of multiple pathogens on host survival exceeds the sum of the effects of each pathogen alone. Thus, many pathogens against which a host has an imperfect immune response can collectively push a host's condition below a threshold where mortality becomes rather high (orange lines in *Figures 3 and 6*). In our model, this assumption is fulfilled rather naturally. Since the probability to survive can logically not exceed 1, the function that maps condition to survival has to be saturating (*Equation 5*).

In the following, we detail how assumptions (a) and (b) can result in the emergence of well over 100 alleles such that heterozygotes have similarly high fitness (condition (1) of Kimura and Crow) and homozygotes have similarly low fitness (condition (2) of Kimura and Crow). We start with the observation that the survival probabilities in evolved polymorphic populations vary between individuals (lower panels in *Figures 4 and 5B–D*). Part of the population consists of individuals that have very low survival probabilities. These are individuals with a condition value considerably less than $K$ and they are almost exclusively homozygotes. This is because, whenever polymorphism is favoured, homozygotes are poorly defended against some pathogens and the fact that pathogens affect condition multiplicatively (*Equation 4*). The remaining part of the population consists of individuals with condition values considerably above $K$. Although the condition of these individuals can differ by several orders of magnitude, their survival is close to 1, which results from the fact that the function that maps condition to survival is saturating. These individuals are almost exclusively heterozygotes. This is because alleles that protect against complementary sets of pathogens, when paired together, offer at least a decent protection against all pathogens. In summary, our assumptions (a) and (b) lead to a set of alleles such that their survival probabilities fall into two clusters as required for conditions (1) and (2) of *Kimura and Crow, 1964* to be fulfilled. The larger the number of alleles, the lower becomes the

proportion of vulnerable homozygotes, and the population consists increasingly of almost equally fit heterozygotes.

*Borghans et al., 2004* use a bit-string model similar to ours with $m = 50$ pathogens, $n_{pep} = 20$ peptides, a virulence of $v = 7$ and a step function for the probability that an MHC-molecule detects a peptide ($a \to \infty$ in *Equation 2*). In contrast to our model, they assume that an individual's condition equals the proportion of detected pathogens, meaning that each pathogen can reduce fitness by only 2% (thereby not fulfilling our assumption a). Additionally, they assume that survival is proportional to the squared condition (not fulfilling our assumption b). *Appendix 6—figure 1* shows a run of our bit-string model with the parameter values used by *Borghans et al., 2004*, resulting in more than 100 coexisting alleles. In contrast, they find only up to seven coexisting alleles, demonstrating that assumptions (a) and (b) in our model drive the high number of coexisting alleles found by us.

Currently, there are several mechanisms proposed to explain the diversity observed at MHC loci. First, in the presence of an HA, each allele has an advantage when rare because it almost always occurs in heterozygotes. Thus, there is negative frequency-dependent selection acting at the level of the allele. In addition, negative frequency-dependent selection can arise from, for example, Red-Queen dynamics, fluctuating selection, and disassortative mating (*Apanius et al., 1997*; *Hedrick, 1998*; *Penn, 2002*; *Borghans et al., 2004*; *Wegner, 2008*; *Spurgin and Richardson, 2010*; *Loiseau et al., 2011*; *Ejsmond and Radwan, 2015*; *Lighten et al., 2017*; *Ejsmond et al., 2023*). These mechanisms are similar to HA in the sense that the selective advantage of an allele increases with decreasing frequency. However, they do not result in heterozygotes being more fit than the homozygotes carrying the rare allele. In addition, neutral diversity can be enhanced by recombination (*Klitz et al., 2012*; *Linnenbrink et al., 2018*; *Robinson et al., 2017*). If many individuals are heterozygous, the particularly high levels of gene conversion found at MHC genes can be effective in creating new allelic variants. For instance, for urban human populations with a large effective population size of $N_e = 10^6$ and a per capita gene conversion probability of $r = 10^{-4}$ an effective number of alleles as high as $n_e = 1 + 4rN_e = 401$ can theoretically be maintained by gene conversion (*Klitz et al., 2012*). However, it is important to point out that for gene conversion to increase allelic diversity, some genetic polymorphism due to balancing selection has to exist to start with. We do not claim that the mechanisms listed here do not play an important role in maintaining allelic diversity at MHC loci. Rather, our results show that, contrary to the currently widespread view, HA should not be dismissed as a potent force. In any real system, different mechanisms will jointly affect allelic diversity. For instance, *Lighten et al., 2017* present a model in which, for Red-Queen co-evolution to maintain allelic polymorphism, HA in the form of a divergent allele advantage (*Wakeland et al., 1990*) seems to be a necessary ingredient. Similarly, *Borghans et al., 2004* show that pathogen co-evolution can further increase the number of coexisting alleles compared to HA alone.

The aim of our study is to understand how HA on its own can result in allelic polymorphism. For this reason, we kept all aspects concerning pathogens fixed, focusing on a scenario where pathogen optima represent diverse taxonomic groups that remain approximately constant over the time scales considered in our model. This approach excludes Red-Queen dynamics and fluctuating selection. Models of Red-Queen dynamics are based that pathogens evolve to avoid detection by the host's immune system (*Borghans et al., 2004*; *Ejsmond and Radwan, 2015*; *Ejsmond et al., 2023*). In our model, this would correspond to moving pathogen optima (in the Gaussian model) or changes in the pathogen peptides (in the bit-string model). We expect that incorporating this would hamper the build-up of allelic MHC diversity when driven solely by HA if pathogens evolve quickly. Alleles previously maintained as beneficial would then become disadvantageous and go extinct more rapidly than new advantageous alleles can appear.

Another component of pathogens that can evolve in response to host immune defence is their virulence (*Frank and Schmid-Hempel, 2008*). The transmission-virulence trade-off hypothesis (*Anderson and May, 1982*; *Frank, 1996*; *Alizon et al., 2009*) predicts that pathogens that cause relatively little harm to their host (i.e. pathogens with low virulence) may evolve towards higher virulence to increase their transmission rate. In line with this hypothesis, we speculate that incorporating virulence evolution leads to higher virulence whenever pathogens inflict little harm on their hosts. This scenario applies in the dark blue parameter regions in *Figures 4 and 5*, where host populations possess a single effective generalist allele. In these regions, the evolution of increased virulence would shift pathogens into parameter regions where allelic polymorphism becomes adaptive. The ensuing build-up of allelic

polymorphism decreases the harm inflicted by pathogens through HA, which, in turn, increases the selection pressure acting on pathogens for an even further increase in virulence. This suggests, in contrast to evolving pathogen optima, a positive feedback loop between virulence evolution and the evolution of allelic diversity.

Our Gaussian model is not restricted to MHC genes, but can apply to any gene that affects several functions important for survival. Examples are genes that are expressed in different ontogenetic stages or different tissues with competing demands on the optimal gene product. However, gene duplication is expected to reduce the potential number of coexisting alleles per locus and eventually lead to a situation where the number of duplicates equals the number of functions (*Proulx and Phillips, 2006*). Under this scenario, the high degree of polymorphism reported here would be transient. However, for MHC genes evidence exist that other forces limit the number of MHC loci (*Penn, 2002*; *Wegner, 2008*; *Eizaguirre and Lenz, 2010*; *Spurgin and Richardson, 2010*). But it is important to point out that, while our model focuses on evolution at a single MHC locus, many vertebrates have more than one MHC locus with similar functions (*Wegner, 2008*; *Eizaguirre and Lenz, 2010*; *Spurgin and Richardson, 2010*). The diversity generating mechanism described here still applies if the different loci are responsible for largely non-overlapping sets of pathogens, indicating that the mechanism presented here can in principle explain the high number of coexisting MHC-alleles.

In summary, our research offers a fresh view that can help us to understand allelic diversity at MHC loci. We identify two crucial assumptions related to pathogen-host interactions, under which we show that HA emerges as a potent force capable of driving the evolution of a very high number of coexisting alleles.

## Acknowledgements

We thank Yvonne Meyer-Lucht and Tobias Lenz for helpful discussions and Göran Arnqvist, Helena Westerdahl, Joachim Hermisson, and Sophie Karrenberg for comments on an earlier version of the manuscript.

## Additional information

### Funding
No external funding was received for this work.

### Author contributions
Mattias Siljestam, Conceptualization, Formal analysis, Visualization, Methodology, Writing – original draft, Writing – review and editing; Claus Rueffler, Conceptualization, Supervision, Funding acquisition, Methodology, Writing – original draft, Writing – review and editing

### Author ORCIDs
Mattias Siljestam ![ORCID] https://orcid.org/0000-0002-3720-4926
Claus Rueffler ![ORCID] https://orcid.org/0000-0001-9836-2752

### Decision letter and Author response
Decision letter https://doi.org/10.7554/eLife.94587.sa1
Author response https://doi.org/10.7554/eLife.94587.sa2

## Additional files

### Supplementary files
MDAR checklist

### Data availability
All data presented and analysed in this study were generated through individual based simulations using Matlab, with code authored by the first author. The corresponding Matlab script is available at Dryad with DOI: https://doi.org/10.5061/dryad.69p8cz98j.

The following dataset was generated:

| Author(s) | Year | Dataset title | Dataset URL | Database and Identifier |
|-----------|------|---------------|-------------|------------------------|
| Siljestam M | 2024 | Heterozygote advantage can explain the extraordinary diversity of immune genes | https://doi.org/10.5061/dryad.69p8cz98j | Dryad Digital Repository, 10.5061/dryad.69p8cz98j |

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

## Appendix 1

### Effective number of alleles

Here, we provide the calculations for the effective number of alleles $n_e$ reported in *Figures 4 and 5*. The effective number of alleles is given by the reciprocal of the population homozygosity $G = \sum f_i^2$, where $f_i$ denotes the frequency of allele $i$ in the population (*Kimura and Crow, 1964*). Under mutation-drift balance, the expected homozygosity is approximated by $1/(1 + 4N\mu)$ (*Gillespie, 2004*), where $N$ is population size and $\mu$ the per capita mutation probability.

Thus, under mutation-drift balance, the expected value of $n_e$ equals $1 + 4N\mu$. For *Figure 4*, where $N = 10^5$ and $\mu = 5 \times 10^{-7}$, the expected value of $n_e$ is 1.2. In *Figure 5*, with $N = 10^5$ and $\mu = 5 \times 10^{-6}$, the expected value for $n_e$ is 3. Hence, $n_e$-values significantly higher than these expectations indicate the presence of alleles maintained by balancing selection.

It is worth noting that when alleles are at equal frequencies $f_i = 1/n$, $n_e$ is equal to $n$. In our model, both conditions (1) and (2) of *Kimura and Crow, 1964*, are approached at evolutionary equilibrium (i.e. heterozygote having similar and high fitness while homozygote having similar and low fitness), as elaborated in the Discussion. As a result, alleles maintained by HA are maintained at roughly similar frequencies. Consequently, $n_e$ gives a good estimate for the number of alleles that coexist in a protected polymorphism due to HA, rather than being maintained in a balance between mutation and drift.

## Appendix 2

## Evolutionary dynamics without diversification

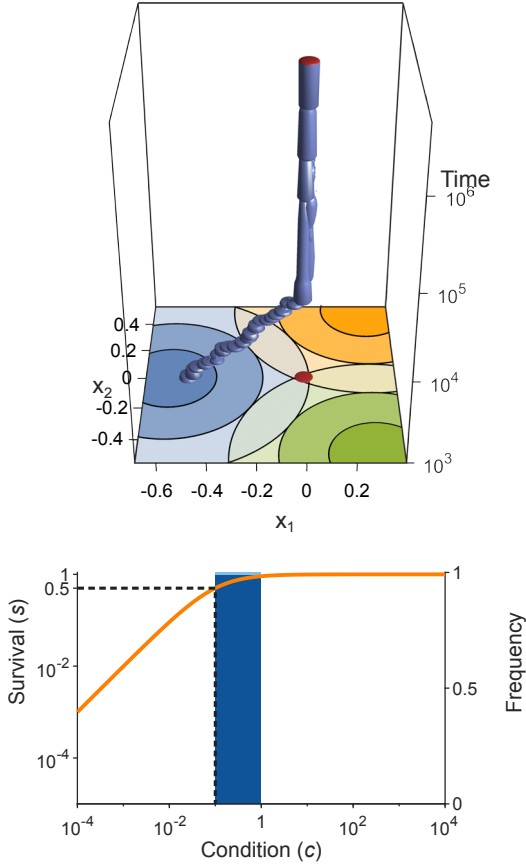

**Appendix 2—figure 1.** Evolution of allelic values in the presence of three pathogens. This figure is analogous to *Figure 3* (see that legend for details) but with wider Gaussians ($v = 2.5$, as in *Figure 1C*). As a consequence, the condition for evolutionary branching ($v > 2\sqrt{m}$) is not fulfilled and the evolutionary dynamics result in a monomorphic population consisting essentially of only the generalist allele $\boldsymbol{x}^* = (0, 0)$. This result is independent of the half-saturation constant $K$, here chosen to be $K = 10$.

# Appendix 3

## Table of mathematical notation

List of all mathematical symbols used in the Supplementary Information. Bold italic font indicates vectors (e.g. $\boldsymbol{x}$) while normal italic font indicates numbers or scalar-valued functions. Capital letters in sans serif font indicate matrices (e.g. H).

| Notation | Explanation |
|---|---|
| $c$ | condition function |
| $c_{\mathrm{max}}$ | maximum condition |
| C | mutational covariance matrix |
| $D$ | peptide detection probability (bit-string model) |
| $\delta$ | expected mutational step size (Gaussian model) |
| $e_k$ | efficiency function for pathogen $k$ |
| $f_a$ | frequency of allele $\boldsymbol{x}_a$ |
| $h$ | dimensionality of the allelic trait space |
| H | Hessian matrix |
| $I$ | identity matrix |
| J | Jacobian matrix |
| $K$ | half-saturation constant of survival function |
| $m$ | number of pathogens |
| $\mu$ | per capita mutation probability |
| $n$ | number of alleles |
| $n_{\mathrm{e}}$ | effective number of alleles |
| $N$ | population size |
| $\boldsymbol{p}_k$ | vector describing the $k$th pathogen (Gaussian model) |
| $s$ | survival function |
| $s_{\mathrm{max}}$ | maximum survival |
| $\Sigma_k$ | covariance matrix of the Gaussian efficiency function $e_k$ for pathogen $k$ (**Equation A14**) |
| $\Sigma_{\mathrm{G}}$ | covariance matrix of the Gaussian efficiency function $e_k$, assuming equal covariance matrices for all pathogens |
| $\sigma_{\mathrm{G}}^2$ | variance of the Gaussian efficiency function $e_k$, assuming full symmetry matrices for all pathogens |
| $\Sigma_p$ | covariance matrix of the position of the pathogen vectors (Gaussian model) |
| $\sigma_p^2$ | variance of the position of the pathogen vectors, assuming equally distant pathogen optima (Gaussian model) |
| $v$ | virulence; given by $\sigma_{\mathrm{G}}^{-1}$ for the Gaussian model and the detection threshold value in the bit-string model |
| $\boldsymbol{x}_a$ | allelic trait vector of allele $a$ |
| $\boldsymbol{x}$ | allelic trait vector of a resident allele |
| $\boldsymbol{y}$ | allelic trait vector of rare mutant allele |

## Appendix 4

### Varying the expected mutational step size in the Gaussian model

For the Gaussian model, mutations are drawn from an isotropic normal distribution, i.e., a matrix with covariance matrix $\sigma_\mu I$ of dimension $h$. The expected mutational step size $\delta$ is given by $\sigma_\mu$ times the expected value of the Chi-distribution (Equation 18.14 in *Johnson et al., 1994*),

$$\delta = \sigma_\mu \sqrt{2} \frac{\Gamma(\frac{h+1}{2})}{\Gamma(h/2)}. \tag{A1}$$

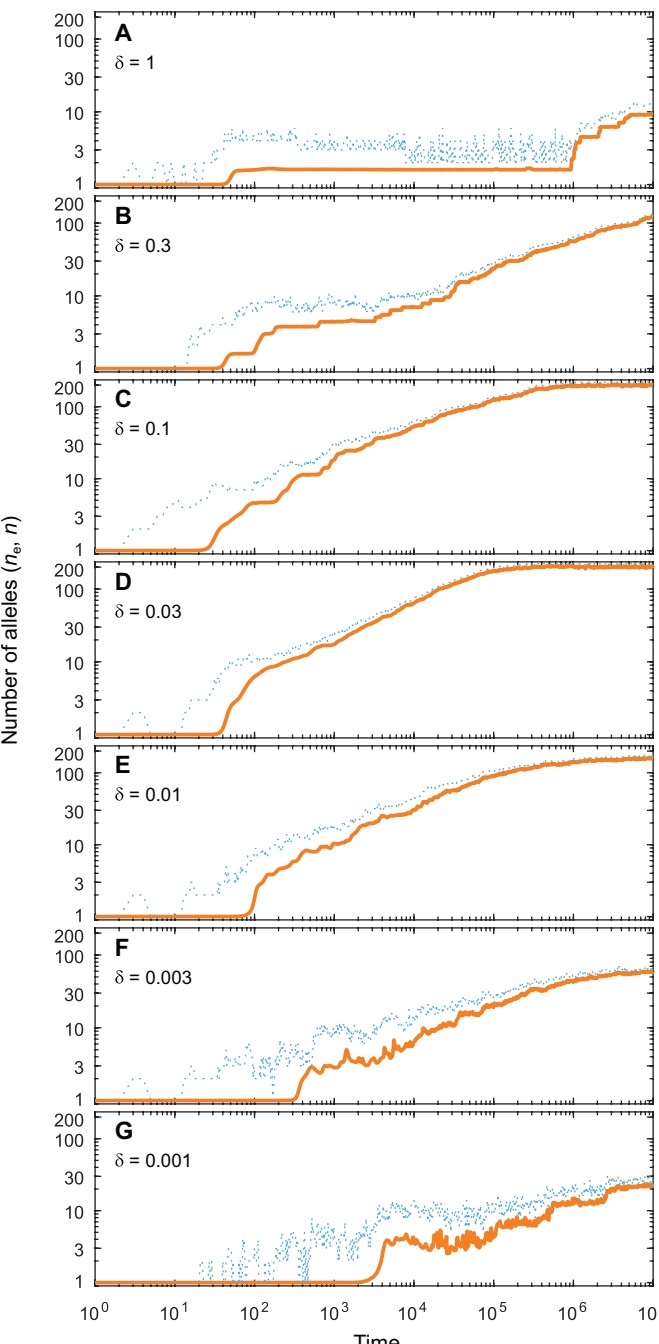

**Appendix 4—figure 1.** Number of coexisting alleles as they emerge in individual-based simulations for different expected mutational step sizes $\delta$ and eight pathogens ($m = 8$). Parameters are chosen such that up to 200 alleles can evolve ($K = 0.5$, $v = 20$; see bottom right panel in *Figure 4* in the main text). Solid orange lines and dotted

blue lines give the effective number $n_e$ and the absolute number $n$ of alleles, respectively. The number of alleles increases fastest and saturates earliest for an intermediate expected mutational step size of $\delta = 0.03$ (**D**; pathogen vectors are $1/\delta = 1/0.03 \approx 30$ average mutational steps apart) as used in **Figure 4**. Decreasing the average mutational step size slows down the build-up of allelic diversity (**E–G**). In the extreme case shown in G (pathogen vectors are 1/0.001 = 1000 average mutational steps apart), the evolutionary dynamics is strongly limited by the rate of phenotypic change due to the small step size and the number of alleles after $10^7$ time steps has reached only 10% the number reached in D. Increasing the average mutational step size also slows down the build-up of allelic diversity (**A–C**). In the extreme case shown in A (pathogen vectors are 1.25 average mutational steps apart), the evolutionary dynamics are strongly limited by the very large proportion of maladapted mutants. Other parameters (as in **Figure 4**): $N = 10^5$, $\mu = 5 \times 10^{-7}$.

## Appendix 5

### Deviations from symmetry in the Gaussian model

The number of coexisting alleles for different parameter combinations are shown in *Figure 4* in the main text. These results are based on two symmetry assumptions. First, the $m$ points describing by the pathogen vectors are placed equidistantly with $d = 1$, resulting in a regular $(m - 1)$-simplex. Second, the multivariate Gaussian functions $e_k$ describing the MHC-molecule's efficiencies against the different pathogens are isotropic and have equal width, as shown in *Figure 1*. Thus, the covariance matrices $\Sigma_k$ in *Equation A14* are equal to $\sigma_G^{-2}I$, where $I$ is the identity matrix. Here, we test the robustness of the outcome shown *Figure 4* with respect to violations of these symmetry assumptions. We focus on the case with eight pathogens, and the results are summarised in *Appendix 5—figure 1*. Panel A is identical to the bottom right panel in *Figure 4*, and shown here for comparison. Panels B–D show the final number of coexisting alleles for increasing deviations from symmetry, as explained in the following. Note that each pixel in the figure is based on a single simulation with a unique random perturbation from symmetry.

In *Appendix 5—figure 1B* the assumption of symmetrically placed pathogen vectors is perturbed while the Gaussian functions $e_k$ are kept rotationally symmetric with equal width. Section 'Random placement of pathogen vectors' describes the procedure how the positions of the pathogen vectors are randomised. The similarity between panels A and B indicates that deviations from a symmetric placement of pathogen vectors has a minor effect on the number of coexisting alleles. The slightly decreased smoothness of the contours corresponding to 50 and 100 coexisting alleles stems from the fact that each simulation (corresponding to a pixel) is based from a unique perturbation. Note that polymorphism can emerge for values of $v$ such that the branching condition $v > 2\sqrt{m}$ derived for the symmetric case is not fulfilled (below the red arrow). This can be understood based on the expression for the Hessian matrix given in *Equation A43*. This Hessian matrix is more likely to be positive definite for asymmetrically placed pathogen vectors.

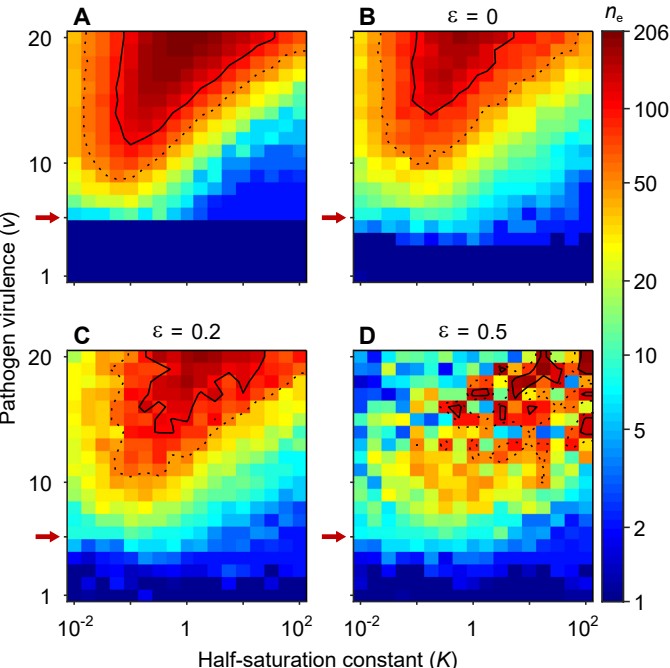

**Appendix 5—figure 1.** Number of coexisting alleles for eight pathogens as a function of pathogen virulence $v$ and the half-saturation constant $K$ for symmetrically (**A**) and non-symmetrically placed pathogen vectors (**B–D**). Figures are based on a single individual-based simulation per pixel and run for $10^6$ generations, assuring that the equilibrium distribution of alleles is reached. (**A**) shows results for equally spaced pathogen vectors and isotropic functions $e_k$ (*Equation A14*). It is identical to the bottom right panel in *Figure 4* and shown here for comparison. (**B–D**) show the result for increasing perturbations from symmetry. In (**B**), pathogen vectors are placed randomly (see Section 'Random placement of pathogen vectors' for details) while the functions $e_k$ are kept rotationally

*Appendix 5—figure 1 continued*

symmetric. In (**C**) and (**D**), additionally to the non-symmetric placement of pathogen vectors, the functions $e_k$ are independently perturbed from rotational symmetry (see Appendix 'Random covariance matrices for the pathogen efficiencies' for details). In (**C**) the deviations from rotational symmetry are moderate, while in (**D**) they are strong. Note that in (**B–D**) pathogen vectors are no longer at a constant distance 1, but instead have the mean variance calculated from the pathogen optima corresponds to the variance of symmetrically placed pathogens optima with distance 1. Results are reported in terms of the effective number of alleles $n_e$, which discounts for alleles arising from mutation-drift balance (see Appendix 'Effective number of alleles'). Dashed and solid lines give the contours for $n_e = 50$ and $n_e = 100$, respectively. Red arrows indicate $v = 2\sqrt{8}$, the minimal value for polymorphism to emerge from branching under full symmetry (**Equation A46**). Accordingly, simulations in the dark blue area result in a single abundant allele with $n_e$ close to one. Other parameters: population size $N = 10^5$, per capita mutation probability $\mu = 5 \times 10^{-7}$, expected mutational step size $\delta = 0.03$.

In **Appendix 5—figure 1C and D** we, additionally to the non-symmetric placement of pathogen vectors, allow for Gaussian functions $e_k$ that are not isotropic. The variances of the perturbed covariance matrices are drawn from the interval $(\sigma_p^2(1 - \varepsilon), \sigma_p^2(1 + \varepsilon))$ and constrained such that the average variance is equal to 1, and then rotated randomly. Section 'Random covariance matrices for the pathogen efficiencies' describes this procedure in detail. Panel C shows the result for modest ($\varepsilon = 0.2$) and panel D for strong ($\varepsilon = 0.5$) deviations from rotational symmetry. Comparing panels C to B indicates that modest deviations from rotational symmetry have a relatively minor effect on the final number of coexisting alleles. In contrast, in panel D configurations exist where significantly fewer alleles are able to coexist. Interestingly, configurations resulting in a high number of alleles are more likely to occur in combination with high $K$-values. The highly irregular pattern results from each pixel corresponding to a single simulation with a unique random perturbation from symmetry. Furthermore, the threshold for polymorphism decreases even more because the Hessian matrix given in **Equation A42** is even more likely to be positive definite with perturbations in $\Sigma_k$.

## Random placement of pathogen vectors

We here describe how we randomly place eight pathogen vectors in trait space. In order to keep the results comparable to the symmetric case, we keep the average variance calculated from the position of their mid-points constant. The distribution of eight pathogen vectors can be described by their seven dimensional covariance matrix $\Sigma_p$ calculated from the coordinates $\boldsymbol{p}_1, \ldots, \boldsymbol{p}_8$. Since each diagonal element of $\Sigma_p$ describes the variance of the pathogen vectors along a different dimension of the trait space, the average variance equals $\text{tr}(\Sigma_p)/7$, where $\text{tr}(\Sigma_p)$ denotes the trace. This measure is unaffected by rotation of the points $\boldsymbol{p}_1, \ldots, \boldsymbol{p}_8$. For symmetrically placed pathogen vectors $\Sigma_{p,\text{sym}} = d^2/(2m)I$, where $I$ denotes the identity matrix, and therefore $\text{tr}(\Sigma_{p,\text{sym}}) = d^2(m - 1)/(2m)$. For the pathogens with perturbed placements (with covariance matrix $\Sigma_{p,\text{per}}$), we demand $\text{tr}(\Sigma_{p,\text{per}}) = \text{tr}(\Sigma_{p,\text{sym}})$. We achieve this by first choosing eight preliminary points $\hat{\boldsymbol{p}}_1, \ldots, \hat{\boldsymbol{p}}_8$ that are placed randomly within a unit 7-sphere, having the covariance matrix $\hat{\Sigma}_p$. By subsequently multiplying the coordinates of these points by a scalar $\alpha$, the variances in $\hat{\Sigma}_p$ are multiplied by $\alpha^2$. By setting

$$\alpha = \sqrt{\frac{\text{tr}(\sum_{p,\text{sym}})}{\text{tr}(\hat{\sum}_p)}} = \sqrt{\frac{d^2(m - 1)}{2m\,\text{tr}(\hat{\sum}_p)}}, \tag{A2}$$

with $m = 8$, we obtain the final set of pathogen vectors $\boldsymbol{p}_1, \ldots, \boldsymbol{p}_8$ with a covariance matrix $\Sigma_{p,\text{per}}$ fulfilling $\text{tr}(\Sigma_{p,\text{per}}) = \text{tr}(\Sigma_{p,\text{sym}})$.

## Random covariance matrices for the pathogen efficiencies

We here describe how we create random covariance matrices $\Sigma_k$. In order to keep the results between the symmetric and asymmetric case comparable, we fix the mean variance over all $\Sigma_k$ to $\sigma^2 = v^{-2}$. We obtain the eight random covariance matrices $\Sigma_1, \ldots, \Sigma_8$ in the following manner. First, eight random diagonal matrices $D_1, \ldots, D_8$ are determined (one per pathogen vectors) with entries drawn from a uniform distribution $U(1 - \varepsilon, 1 + \varepsilon)$. These matrices are then multiplied with the scalar

$$\beta = v^{-2} \frac{m-1}{\frac{1}{m} \sum_{k=1}^{m} \text{tr}(D_k)},$$ (A3)

with $m = 8$ to obtain the set of matrices $M_1, \ldots, M_8$ obeying $v^{-2} = \frac{1}{8} \sum_{i=1}^{8} (\text{tr}(M_i)/7)$. In a final step, we draw eight random rotation matrices $R_1, \ldots, R_8$ and calculate our final covariance matrices $P_1, \ldots, P_8$ as $P_k = R_k M_k R_k^{\text{T}}$.

## Appendix 6

## Simulation run of the bit-string model with parameter values as in *Borghans et al., 2004*

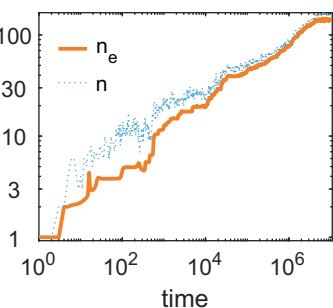

**Appendix 6—figure 1.** The number of alleles $n$ and the effective number of alleles $n_e$ as a function of time (on a log-log plot) for a simulation run of our bit-string model. Parameters values: $N = 10^5$ and $\mu = 5 \times 10^{-6}$. Other parameters as in *Borghans et al., 2004*: $v = 7$, $m = 50$, $n_{\text{pep}} = 20$.

We here present a comparison of our bit-string model with that of *Borghans et al., 2004*. These authors analyse a bit-string model with $m = 50$ pathogens (that are allowed to mutate but are not subject to selection), with $n_{\text{pep}} = 20$ peptides each, a virulence of $v = 7$ (with a step function for the probability that an MHC-molecule detects a peptide), a population size of $N = 10^3$, and a per capita mutation probability of $\mu = 10^{-5}$. In contrast to our model, they assume that condition equals the proportion of detected pathogens, such that each pathogen can lower fitness by only 2% (not fulfilling our assumption a) and that survival is proportional to the squared condition (not fulfilling our assumption b). With these parameters and parameter values, their simulation results in up to seven alleles. We note, that the effective number of alleles in these simulations is likely lower, but no allele frequencies are given.

We contrast their results with those from our model, which, as detailed in the main part, fulfils assumptions (a) and (b). To approximate the step function for the detection probability, we use

$$D(L_{ki}(\boldsymbol{x})) = \frac{1}{1 + \exp[2 \log(99)(v - L_{ki}(\boldsymbol{x}) - 1/2)]}. \tag{A4}$$

For this function, $v$ is the required match length $L$ for a 99% chance of detection, while a match length $L = v - 1$ gives only 1% detection probability. Note, that compared to *Equation 2*, we here subtract 1/2 in the denominator and $a = 2 \log(99)$. Then, our model with the exact same parameters (omitting pathogen mutations) results in 18 alleles and $n_e = 16.7$, clearly exceeding the number of alleles found by *Borghans et al., 2004*.

Based on *Kimura and Crow, 1964*, for the above $N$ and $\mu$ the effective number of alleles that can be maintained by HA cannot exceed $n_e = 17.6$ at mutation-drift-selection balance. This suggests that the allelic diversity found by *Borghans et al., 2004*, is likely not limited by the parameters affecting mutation and drift, $\mu$ and $N$. In contrast, our final number of alleles (being 95% of the maximum) is likely limited by these parameters. To demonstrate that this is indeed the case, we simulate our model with $N = 10^5$ and $\mu = 5 \times 10^{-6}$, as shown in *Appendix 6—figure 1*. We find well over 100 alleles ($n = 157$ and $n_e = 140$). This demonstrates that the ecological parameter values used by *Borghans et al., 2004*, $m = 50$, $n_{\text{pep}} = 20$, and $v = 7$, under our model allows for more than a 20-fold higher allelic diversity.

## Appendix 7

## Mathematical analysis of the Gaussian model: preliminaries

### Adaptive dynamics and invasion fitness

For the Gaussian model presented in the main part, we investigate with an evolutionary invasion analysis using the adaptive dynamics formalism (*Metz et al., 1992*; *Dieckmann and Law, 1996*; *Geritz et al., 1998*) whether selection favours a single generalist allele or a polymorphic population. In the language of adaptive dynamics, we ask whether a monomorphic population evolves towards an evolutionary branching point, where two coexisting allelic lineages emerge.

Let us consider a large population of $N$ individuals with two segregating alleles $x_1$ and $x_2$ under Wright-Fisher population dynamics (*Fisher, 1930*; *Wright, 1931*). The allelic frequencies at time $t$ are denoted $f_{x_1,t}$ and $f_{x_2,t}$, respectively.

The recurrence equation for the change of frequency of an allele $x_a \in \{x_1, x_2\}$ is then given by

$$f_{x_a,t+1} = f_{x_a,t} \left( f_{x_a,t} \frac{s(x_a, x_a)}{\bar{s}_t} + f_{x_b,t} \frac{s(x_a, x_b)}{\bar{s}_t} \right),$$ (A5)

where $s(x_a, x_b)$ is the survival of an individual carrying the alleles $x_a$ and $x_b$ (see *Equation A12*) and

$$\bar{s}_t = f_{x_1,t}^2 s(x_1, x_1) + f_{x_1,t} f_{x_1,t} s(x_1, x_2) + f_{x_2,t}^2 s(x_2, x_2)$$

is the population mean survival at time $t$. Note, that the expression within brackets on the right-hand side of *Equation A5* describes the marginal fitness of allele $x_a$.

Consider a resident population carrying allele $x$ to which a mutant allele $y = x + \epsilon$ is introduced. In the limit of a mutant allele frequency close to zero, its marginal fitness is given by

$$w(y, x) = \frac{s(y, x)}{s(x, x)}.$$ (A6)

We refer to $w(y, x)$ as invasion fitness, which is the expected long-term exponential growth rate of an infinitesimally rare mutant allele $y$ in a resident population with allele $x$ (*Metz et al., 1992*; *Metz, 2008*). Allele $y$ has a positive probability to invade and increase in frequency if $w(y, x) > 1$ and disappears otherwise.

We denote the gradient of invasion fitness with respect to the mutant allele $y$, evaluated at $y = x$, with $\nabla w(x, x)$. It has the entries

$$\nabla w(x, x)_i = \frac{\partial w(y, x)}{\partial y_i} \bigg|_{y=x}$$ (A7)

and gives the direction in the $h$-dimensional allelic trait space in which deviations from $x$ result in the fastest increase of invasion fitness.

If mutations rarely occur, a mutant allele $y$ will either go extinct or reach an equilibrium frequency before the next mutant appears. If, additionally, $\nabla w(x, x) \neq 0$ and mutational effects are sufficiently small (i.e. $y = x + \epsilon$ for $\epsilon$ small), then invasion of $y$ implies extinction of $x$ (*Dercole and Rinaldi, 2008*; *Priklopil and Lehmann, 2020*).

In the limit of small mutational steps, the evolutionary dynamics of an allelic lineage becomes gradual and is given by

$$\frac{dx}{dt} = \mu N C \nabla w(x, x)$$ (A8)

(*Dieckmann and Law, 1996*; *Champagnat et al., 2006*; *Durinx et al., 2008*; *Metz and de Kovel, 2013*). Here, $\mu$ is the per capita mutation probability and C the covariance matrix for the distribution of mutational effects on the trait $x$.

We note that *Equation A8* is structurally similar to the gradient equation of quantitative genetics, which is based on the assumption of weak selection or, equivalently, small genetic variances (*Lande, 1979*; *Iwasa et al., 1991*; *Abrams et al., 1993*; *Débarre et al., 2014*). In this case, $x$ characterises the mean of the phenotype distribution, the covariance matrix describes the distribution of the standing genetic variation, and the factor $\mu N$ is replaced with a constant.

Allelic trait values $\boldsymbol{x}$ where $\nabla w(\boldsymbol{x}, \boldsymbol{x}) = 0$ are of special interest, and such $\boldsymbol{x}$ are referred to as evolutionarily singular points $\boldsymbol{x}^*$.

Evolutionarily singular points can be either attractors or repellers of the evolutionary dynamics described by *Equation A8*. Furthermore, an evolutionarily singular point can be either invadable or uninvadable by nearby mutants. For a resident allele with a one-dimensional trait $\boldsymbol{x} = x$, a classification of singular strategies is straightforward (*Geritz et al., 1998*). Evolutionarily singular points that are not approached, irrespective of whether they are invadable or uninvadable, act as repellers, and we do not expect to ever find resident alleles with such values. Evolutionarily singular strategies that are attractors and uninvadable are endpoints of the evolutionary dynamics. Finally, evolutionarily singular points that are attractors and invadable are known as evolutionary branching points. In this case, any nearby mutant can invade the singular point and coexist with it in a protected dimorphism. Further evolution leads to divergence of the alleles present in the dimorphism. Thus, evolutionary branching points are points in trait space at which diversity emerges (*Geritz et al., 1998*; *Rueffler et al., 2006*).

The classification of singular points becomes more complicated in multivariate trait spaces or when several strategies coexist in an evolutionarily singular point (*Leimar, 2009*; *Doebeli, 2011*; *Geritz et al., 2016*). First, in multivariate trait spaces or polymorphic populations, whether a singular point is an attractor does not only depend on the direction of the fitness gradient in the vicinity of the singular point but also on the mutational input (*Leimar, 2009*). Second, in multivariate trait spaces or polymorphic populations, for evolutionary branching it is necessary that a singular point is an attractor and invadable. However, in the multidimensional case, this is generally not sufficient any more (*Geritz et al., 2016*).

In Section 'The evolutionarily singular point', we show for our model that a unique singular point $\boldsymbol{x}^*$ exists. This allele is uninvadable if it is a minimum of $w(\boldsymbol{y}, \boldsymbol{x}^*)$ as a function of $\boldsymbol{y}$. This is the case if the $h$-dimensional Hessian matrix H with entries

$$h_{ij} = \frac{\partial^2 w(\boldsymbol{y}, \boldsymbol{x})}{\partial y_i \partial y_j} \Big|_{y=x=x^*} .\tag{A9}$$

is negative definite (*Leimar, 2009*; *Doebeli, 2011*). In Section 'Derivation of the Hessian matrix of invasion fitness' we derive an explicit expression for H for the fully general case of our model that allows to determine invadability of $\boldsymbol{x}^*$ as a function of the positions of the pathogen vectors, the half-saturation constant $K$, and the covariance matrices $\Sigma_k$ that determine the shape of the efficiency functions $e_k$.

Whether the singular point $\boldsymbol{x}^*$ is an attractor of the evolutionary dynamics can be evaluated based on the Jacobian matrix J of the fitness gradient $\nabla w(\boldsymbol{x}^*, \boldsymbol{x}^*)$ (*Leimar, 2009*), which is given by

$$J = H + Q \tag{A10}$$

and where Q is the $h$-dimensional matrix of mixed derivatives with entries

$$q_{ij} = \frac{\partial^2 w(\boldsymbol{y}, \boldsymbol{x})}{\partial y_i \partial x_j} \Big|_{y=x=x^*} .\tag{A11}$$

*Leimar, 2009*, shows that if the symmetric part of J, i.e., $(J + J^T)/2$, is negative definite, then the singular point is an attractor of the evolutionary dynamics described by *Equation A8* independent of the mutational covariance matrix C and he refers to this case as strong convergence stability. For the case that the Jacobian matrix is a symmetric negative definite matrix, a stronger result holds, to which he refers to as absolute convergence stability (*Leimar, 2002*; *Leimar, 2009*). In this case, all conceivable gradualistic, adaptive paths starting near the point $\boldsymbol{x}^*$ converge to it. Furthermore, he shows that the condition for absolute convergence stability is equivalent to the existence of a function $g(\boldsymbol{x})$ having a maximum at $\boldsymbol{x}^*$ and a positive function $\alpha(\boldsymbol{x})$ such that the gradient of invasion fitness can be expressed as

$$\nabla w(x, x) = \alpha(x) \nabla g(x).$$

In Section 'Absolute convergence stability', we show for our model that $w(\boldsymbol{y}, \boldsymbol{x}) > 1$ is indeed absolutely convergence stable.

For the case of two-dimensional trait spaces, results in *Geritz et al., 2016*, allow us to conclude that if $x^*$ is invadable, then it is indeed an evolutionary branching point. For trait spaces of dimension three or higher, whether convergence stability and invadability imply evolutionary branching is an open problem (*Geritz et al., 2016*). Individual-based simulations indicate, however, that for our model this is indeed the case.

## Model description

In this section, we describe the model ingredients. Survival $s(x_a, x_b)$ of a genotype carrying alleles $x_a$ and $x_b$ is a saturating function of condition $c$ and described by the well-known Michaelis-Menten equation

$$s(x_a, x_b) = \frac{s_{\max} c(x_a, x_b)}{K + c(x_a, x_b)}.$$ (A12)

Here, the half-saturation constant $K$ gives the condition $c$ at which half of the maximum survival is reached and $s_{\max}$ is the maximum survival probability that is approached when $c$ becomes large.

The condition of a genotype is given by

$$c(x_a, x_b) = c_{\max} \prod_{k=1}^{m} \frac{e_k(x_a) + e_k(x_b)}{2},$$ (A13)

where $c_{\max}$ is the condition of a hypothetical individual with perfect defence against all $m$ pathogens and $e_k(x)$ is the efficiency of an allele's MHC-molecule against pathogen $k$ in an environment with $m$ pathogens.

Without loss of generality, $c(x^*, x^*)$ is standardised to 1 (by choosing $c_{\max}$ in *Equation A13* appropriately). This is helpful because it allows us to choose an interval of $K$-values where individuals homozygous for the generalist allele have either a condition in the range where survival changes rapidly ($K \gg 1$) or slowly ($K \ll 1$) with condition. In *Figure 4*, *Appendix 5—figure 1*, the x-axis can be translated into survival $s(x^*, x^*)$ of the generalist genotype using *Equation A12*, which then varies between 0.01 for $K = 10^{-2}$ and 0.99 for $K = 10^2$.

We assume that the efficiencies $e_k(x)$ of inducing immune defence against the $m$ different pathogens are traded off. This trade-off emerges by describing the efficiencies against different pathogens with multivariate Gaussian functions (see *Figure 1*) that have pathogen-dependent optima,

$$e_k(x) = \exp\left(-\frac{1}{2}(x - p_k)^{\mathrm{T}} \sum_k^{-1} (x - p_k)\right).$$ (A14)

These function describes how the efficiency of an allele characterised by the $h$-dimensional vector $x$ decreases with increasing distance from the pathogen vector $p_k$. The closer an allelic trait vector is to a pathogen vector, the higher is the efficiency of the MHC-molecule against that pathogen. The magnitude of the decrease in efficiency with increasing distance to the $k$th pathogen is determined by the shape and width of the Gaussian function as determined by the $h$-dimensional covariance matrix $\Sigma_k$.

In the main part, we consider the special case of rotationally symmetric Gaussian functions $e_k(x)$. These matrices are thus specified by an inverse matrix-covariance matrix $\Sigma_k^{-1}$ (see *Equation A14*) that takes the form of a scalar matrix, i.e., a scalar multiple of the identity matrix $I$. Furthermore, we assume that all Gaussians are of equal width. Hence, we have a common scalar for all Gaussians that we denote with $v^2$, i.e., $v$ is the inverse of the width of the Gaussian function. We refer to $v$ as virulence (see Section 'Special case: maximal symmetry', below).

$$e_k(x) = \exp\left(-\frac{v^2}{2}(x - p_k)^{\mathrm{T}}(x - p_k)\right)$$ (A15)

# Appendix 8

## Analytical results for the Gaussian model
### The evolutionarily singular point
In this section, we analyse the evolutionary dynamics of a monomorphic resident population in full generality. By subsequently applying several symmetry assumptions, we then derive the analytical results presented in the main text (see Sections 'Special case: identically shaped Gaussian efficiency functions' and 'Special case: maximal symmetry'). Invasion fitness of a rare mutant allele $\boldsymbol{y}$ in a resident population with allele $\boldsymbol{x}$ is given by its marginal fitness,

$$w(\boldsymbol{y}, \boldsymbol{x}) = \frac{s(\boldsymbol{y}, \boldsymbol{x})}{s(\boldsymbol{x}, \boldsymbol{x})} \tag{A16}$$

(see derivation of **Equation A6**). Note, that $s_{\max}$ cancels out. It is therefore omitted from all further calculations. The direction of the evolutionary dynamics is governed by the selection gradient. Its $i$th entry calculates to

$$\nabla w(\boldsymbol{x}, \boldsymbol{x})_i = \frac{\partial w(\boldsymbol{y}, \boldsymbol{x})}{\partial y_i}\bigg|_{y=x} = \frac{1}{s(\boldsymbol{x}, \boldsymbol{x})} \frac{\partial s(\boldsymbol{y}, \boldsymbol{x})}{\partial y_i}\bigg|_{y=x}. \tag{A17}$$

Using the definitions for $s$ (**Equation A12**) and $c$ (**Equation A13**) and their derivatives,

$$\frac{\partial s(\boldsymbol{y}, \boldsymbol{x})}{\partial y_i} = \frac{K}{\left(K + c(\boldsymbol{y}, \boldsymbol{x})\right)^2} \frac{\partial c(\boldsymbol{y}, \boldsymbol{x})}{\partial y_i} \tag{A18}$$

and

$$\frac{\partial c(\boldsymbol{y}, \boldsymbol{x})}{\partial y_i} = c(\boldsymbol{y}, \boldsymbol{x}) \sum_{k=1}^{m} \frac{1}{e_k(\boldsymbol{y}) + e_k(\boldsymbol{x})} \frac{\partial e_k(\boldsymbol{y})}{\partial y_i}, \tag{A19}$$

where **Equation A19** is obtained by applying the generalised product rule

$$\frac{\partial}{\partial \boldsymbol{x}} \left( \prod_{i=1}^{n} f_i(\boldsymbol{x}) \right) = \left( \prod_{i=1}^{n} f_i(\boldsymbol{x}) \right) \sum_{i=1}^{n} \frac{f_i'(\boldsymbol{x})}{f_i(\boldsymbol{x})}, \tag{A20}$$

we obtain

$$\nabla w(\boldsymbol{x}, \boldsymbol{x})_i = \frac{K}{c(\boldsymbol{x}, \boldsymbol{x})\left(K + c(\boldsymbol{x}, \boldsymbol{x})\right)} \frac{\partial c(\boldsymbol{y}, \boldsymbol{x})}{\partial y_i}\bigg|_{y=x} = \frac{K}{2\left(K + c(\boldsymbol{x}, \boldsymbol{x})\right)} \sum_{k=1}^{m} \frac{1}{e_k(\boldsymbol{x})} \frac{\partial e_k(\boldsymbol{x})}{\partial x_i}. \tag{A21}$$

In the next step, we calculate the derivative of the function $e_k(\boldsymbol{x})$ (**Equation A14**). Applying the chain rule and simplifying results in

$$\begin{aligned}
\frac{\partial e_k(\boldsymbol{x})}{\partial x_i} &= -\frac{1}{2} e_k(\boldsymbol{x}) \frac{\partial}{\partial x_i} \left( (\boldsymbol{x} - \boldsymbol{p}_k)^{\mathrm{T}} \Sigma_k^{-1} (\boldsymbol{x} - \boldsymbol{p}_k) \right) \\
&= -\frac{1}{2} e_k(\boldsymbol{x}) \frac{\partial}{\partial x_i} \sum_{j=1}^{h} \sum_{l=1}^{h} \sigma_{kjl}^{-1} (x_j - p_{kj})(x_l - p_{kl}) \\
&= -\frac{1}{2} e_k(\boldsymbol{x}) \left( \sum_{j=1}^{h} \sum_{l=1}^{h} \sigma_{kjl}^{-1} (x_l - p_{kl}) \frac{\mathrm{d}x_j}{\mathrm{d}x_i} + \sum_{j=1}^{h} \sum_{l=1}^{h} \sigma_{kjl}^{-1} (x_j - p_{kj}) \frac{\mathrm{d}x_l}{\mathrm{d}x_i} \right),
\end{aligned} \tag{A22}$$

where the entries of the matrix $\Sigma_k^{-1}$ are denoted by $\sigma_{kjl}^{-1}$. Using that $\mathrm{d}x_j/\mathrm{d}x_i = 0$ for $i \neq j$ and $\mathrm{d}x_j/\mathrm{d}x_i = 1$ for $i = j$ this further simplifies to

$$\frac{\partial e_k(\boldsymbol{x})}{\partial x_i} = e_k(\boldsymbol{x}) \sum_{j=1}^{h} \sigma_{kij}^{-1} (p_{kj} - x_j). \tag{A23}$$

Substituting **Equation A23** into **Equation A21** finally results in

$$\nabla w(\boldsymbol{x}, \boldsymbol{x})_i = \frac{K}{2(K + c(\boldsymbol{x}, \boldsymbol{x}))} \sum_{k=1}^{m} \sum_{j=1}^{h} \sigma_{kij}^{-1}(p_{kj} - x_j) \tag{A24}$$

and

$$\nabla \boldsymbol{w}(\boldsymbol{x}, \boldsymbol{x}) = \frac{K}{2(K + c(\boldsymbol{x}, \boldsymbol{x}))} \sum_{k=1}^{m} \Sigma_k^{-1}(\boldsymbol{p}_k - \boldsymbol{x}). \tag{A25}$$

As mentioned in Section 'Adaptive dynamics and invasion fitness', singular points $\boldsymbol{x}^*$ are allelic trait vectors for which *Equation A25* equals zero. From *Equation A25* follows that in our model singular points have to fulfil

$$\Big( \sum_{k=1}^{m} \Sigma_k^{-1} \Big) \boldsymbol{x}^* = \sum_{k=1}^{m} \Sigma_k^{-1} \boldsymbol{p}_k. \tag{A26}$$

Solving for $\boldsymbol{x}^*$ yields

$$\boldsymbol{x}^* = \Big( \sum_{k=1}^{m} \Sigma_k^{-1} \Big)^{-1} \sum_{k=1}^{m} \Sigma_k^{-1} \boldsymbol{p}_k := \bar{\boldsymbol{p}}_{\mathrm{w}}. \tag{A27}$$

Thus, the unique singular point $\boldsymbol{x}^*$ equals the arithmetic mean of the pathogen vectors $\boldsymbol{p}_1, .. \boldsymbol{p}_m$, each weighted by the inverse of their Gaussian covariance matrices $\Sigma_1, \ldots, \Sigma_m$. For a one-dimensional trait space ($h = 1$) this simplifies to

$$x^* = \frac{\sum_{k=1}^{m} \sigma_k^{-2} p_k}{\sum_{k=1}^{m} \sigma_k^{-2}} =: \bar{p}_{\mathrm{w}}, \tag{A28}$$

which is the well-known weighted average for scalars. If $\Sigma_1 = \ldots = \Sigma_m$, then *Equation A27* simplifies to the arithmetic mean pathogen vector

$$\boldsymbol{x}^* = \frac{1}{m} \sum_{k=1}^{m} \boldsymbol{p}_k := \bar{\boldsymbol{p}} \tag{A29}$$

as stated in the main text.

## Absolute convergence stability

Below, we prove that the unique singular point $\boldsymbol{x}^*$ (*Equation A27*) is absolutely convergence stable. To this end, we first demonstrate that the gradient of invasion fitness can be expressed as $\nabla w(x, x) = \alpha(x) \nabla c(x, x)$, where $\alpha(x)$ is a positive function, and then show that $\boldsymbol{x}^*$ is a maximum of $c(x, x)$.

### Gradient of condition

An individual homozygote for $\boldsymbol{x}$ has a condition given by

$$c(\boldsymbol{x}, \boldsymbol{x}) = c_{\max} \prod_{k=1}^{m} e_k(\boldsymbol{x}), \tag{A30}$$

and the $i$th entry of the gradient $\nabla c(x, x)$ is given by

$$\nabla c(x, x)_i = c(\boldsymbol{x}, \boldsymbol{x}) x_i = c(\boldsymbol{x}, \boldsymbol{x}) \sum_{k=1}^{m} \frac{1}{e_k(\boldsymbol{x})} e_k(\boldsymbol{x}) x_i. \tag{A31}$$

Substituting *Equation A23* into *Equation A31* gives

$$\nabla c(x, x)_i = c(\boldsymbol{x}, \boldsymbol{x}) \sum_{k=1}^{m} \sum_{j=1}^{h} \sigma_{kij}^{-1}(p_{kj} - x_j), \tag{A32}$$

and

$$\nabla c(x,x) = c(\boldsymbol{x},\boldsymbol{x}) \sum_{k=1}^{m} \sum_{j=1}^{h} \Sigma_k^{-1}(\boldsymbol{p}_k - \boldsymbol{x}). \tag{A33}$$

By comparing *Equation A33* with *Equation A25* we see that

$$\nabla \boldsymbol{w}(\boldsymbol{x},\boldsymbol{x}) = \frac{K}{2(K + c(\boldsymbol{x},\boldsymbol{x}))} \sum_{k=1}^{m} \Sigma_k^{-1}(\boldsymbol{p}_k - \boldsymbol{x}) = \frac{K}{2c(\boldsymbol{x},\boldsymbol{x})(K + c(\boldsymbol{x},\boldsymbol{x}))} \nabla c(x,x), \tag{A34}$$

where the fraction before $\nabla c(x,x)$ is positive. Thus, $\nabla w(x,x) = \alpha(x)\nabla c(x,x)$.

## Hessian matrix of condition

The Hessian matrix of a homozygote individual's condition, evaluated at the singular point, is given by the second-order partial derivative of $c(\boldsymbol{x}^*,\boldsymbol{x}^*)$ with respect to the $i$th and $j$th entry of $\boldsymbol{x}^*$.

We obtain this by differentiating *Equation A33* with respect of $x_j$, evaluated at $x = x^*$, resulting in

$$h_{cij} = \frac{\partial^2 c(\boldsymbol{x}^*,\boldsymbol{x}^*)}{\partial x_i^* x_j^*} = \frac{\partial}{\partial x_j^*}\left(c(\boldsymbol{x}^*,\boldsymbol{x}^*) \sum_{k=1}^{m} \sum_{l=1}^{h} \sigma_{kil}^{-1}(p_{kl} - x_l^*)\right), \tag{A35}$$

and applying the product rule and using that the first derivative at $\boldsymbol{x}^*$ is zero gives

$$h_{cij} = c(\boldsymbol{x}^*,\boldsymbol{x}^*) \sum_{k=1}^{m} \sum_{l=1}^{h} \sigma_{kil}^{-1} \frac{\partial(p_{kl} - x_l^*)}{\partial x_j^*} = -c(\boldsymbol{x}^*,\boldsymbol{x}^*) \sum_{k=1}^{m} \sigma_{kil}^{-1}, \tag{A36}$$

where the last simplification uses that $dx_j/dx_i = 0$ for $i \neq j$ and $dx_j/dx_i = 1$ for $i = j$, and

$$H_c = -c(\boldsymbol{x}^*,\boldsymbol{x}^*) \sum_{k=1}^{m} \Sigma_k^{-1}, \tag{A37}$$

which is always negative definite. Thus, $c(\boldsymbol{x}^*,\boldsymbol{x}^*)$ is a maximum and $\boldsymbol{x}^*$ is absolute convergence stable.

## Derivation of the Hessian matrix of invasion fitness

As stated in *Equation A9*, the entries of the Hessian matrix of invasion fitness are given by

$$h_{ij} = w(\boldsymbol{y},\boldsymbol{x}^*)y_iy_j = \frac{1}{s(\boldsymbol{x}^*,\boldsymbol{x}^*)}s(\boldsymbol{y},\boldsymbol{x}^*)y_iy_j. \tag{A38}$$

The second derivative of the function $s$ is obtained by differentiating *Equation A18* with respect to $y_j$, resulting in

$$
\begin{aligned}
\frac{\partial^2 s(\boldsymbol{y},\boldsymbol{x}^*)}{\partial y_iy_j}\bigg|_{y=x^*} &= K\frac{\partial}{\partial y_j}\left(\frac{1}{(K + c(\boldsymbol{y},\boldsymbol{x}))^2}\frac{\partial c(\boldsymbol{y},\boldsymbol{x})}{\partial y_i}\right)\bigg|_{y=x^*} \\
&= \frac{K}{(K + c(\boldsymbol{x}^*,\boldsymbol{x}^*))^4}\left(\frac{\partial^2 c(\boldsymbol{y},\boldsymbol{x}^*)}{\partial y_iy_j}\bigg|_{y=x^*}(K + c(\boldsymbol{x}^*,\boldsymbol{x}^*))^2 - \frac{\partial c(\boldsymbol{y},\boldsymbol{x}^*)}{\partial y_i}\bigg|_{y=x^*}\frac{\partial c(\boldsymbol{y},\boldsymbol{x}^*)^2}{\partial y_j}\bigg|_{y=} \right) \\
&= \frac{K}{(K + c(\boldsymbol{x}^*,\boldsymbol{x}^*))^2}\frac{\partial^2 c(\boldsymbol{y},\boldsymbol{x}^*)}{\partial y_iy_j}\bigg|_{y=x^*}.
\end{aligned}
\tag{A39}
$$

In the final simplification step we use the conclusion drawn from *Equation A21* that $0 = \partial c(\boldsymbol{y},\boldsymbol{x}^*)/\partial y_i|_{y=x^*}$ and therefore the term after the minus sign disappears.

The second derivative for the function $c$ is obtained by differentiating *Equation A19* with respect to $y_j$, resulting in

$$
\begin{aligned}
\frac{\partial^2 c(\boldsymbol{y}, \boldsymbol{x}^*)}{\partial y_i y_j} \Big|_{y=x^*} &= \frac{\partial}{\partial y_j}\left( c(\boldsymbol{y}, \boldsymbol{x}^*) \sum_{k=1}^{m} \left( \frac{1}{e_k(\boldsymbol{y}) + e_k(\boldsymbol{x}^*)} \frac{\partial e_k(\boldsymbol{y})}{\partial y_i} \right)\right) \Big|_{y=x^*} \\
&= c(\boldsymbol{x}^*, \boldsymbol{x}^*) \sum_{k=1}^{m} \frac{\partial}{\partial y_j}\left( \frac{1}{e_k(\boldsymbol{y}) + e_k(\boldsymbol{x}^*)} \frac{\partial e_k(\boldsymbol{y})}{\partial y_i} \right) \Big|_{y=x^*} \\
&= c(\boldsymbol{x}^*, \boldsymbol{x}^*) \sum_{k=1}^{m} \frac{1}{4 e_k(\boldsymbol{x}^*)^2} \left( 2 \frac{\partial^2 e_k(\boldsymbol{y})}{\partial y_i \partial y_j} \Big|_{y=x^*} e_k(\boldsymbol{x}^*) - \frac{\partial e_k(\boldsymbol{y})}{\partial y_i}\Big|_{y=x^*} \frac{\partial e_k(\boldsymbol{y})}{\partial y_j}\Big|_{y=x^*} \right).
\end{aligned} \tag{A40}
$$

Here, the one but last simplification step again follows from the fact that $0 = \partial c(\boldsymbol{y}, \boldsymbol{x}^*)/\partial y_i|_{\boldsymbol{y}=\boldsymbol{x}^*}$. The second derivative for the function $e_k$ is obtained by differentiating *Equation A23* with respect to $y_j$, resulting in

$$
\begin{aligned}
\frac{\partial^2 e_k(\boldsymbol{y})}{\partial y_i \partial y_j} &= \frac{\partial}{\partial y_j} e_k(\boldsymbol{y}) \sum_{l=1}^{h} \sigma_{kil}^{-1}(p_{kl} - y_l) \\
&= \frac{\partial e_k(\boldsymbol{y})}{\partial y_j} \sum_{l=1}^{h} \sigma_{kil}^{-1}(p_{kl} - y_l) - e_k(\boldsymbol{y})\left( \sum_{l=1}^{h} \sigma_{kil}^{-1} \frac{\partial y_l}{\partial y_j} \right) \\
&= \frac{1}{e_k(\boldsymbol{y})} \frac{\partial e_k(\boldsymbol{y})}{y_i} \frac{\partial e_k(\boldsymbol{y})}{y_j} - e_k(\boldsymbol{y})\sigma_{kij}^{-1},
\end{aligned} \tag{A41}
$$

where the last simplification uses that $dy_j/dy_i = 0$ for $i \neq j$ and $dy_j/dy_i = 1$ for $i = j$.

By recursively substituting *Equations A39–A41* into *Equation A9* we obtain

$$
\begin{aligned}
h_{ij} &= \frac{K}{K + c(\boldsymbol{x}^*, \boldsymbol{x}^*)} \sum_{k=1}^{m} \frac{1}{4 e_k(\boldsymbol{x}^*)^2} \left( \frac{\partial e_k(\boldsymbol{y})}{\partial y_i}\Big|_{y=x^*} \frac{\partial e_k(\boldsymbol{y})}{\partial y_j}\Big|_{y=x^*} - 2 e_k(\boldsymbol{x}^*)^2 \sigma_{kij}^{-1} \right) \\
&= \frac{K}{4\left(K + c(\boldsymbol{x}^*, \boldsymbol{x}^*)\right)} \sum_{k=1}^{m} \left( \left( \sum_{l=1}^{h} \sigma_{k\,il}^{-1}(p_{kl} - x_l^*) \right)\left( \sum_{l=1}^{h} \sigma_{k\,jl}^{-1}(p_{kl} - x_l^*) \right) - 2\sigma_{k\,ij}^{-1} \right),
\end{aligned}
$$

where in the last step we substituted *Equation A23*. This result can be rewritten as a matrix

$$
\mathsf{H} = \frac{K}{4\left(K + c(\boldsymbol{x}^*, \boldsymbol{x}^*)\right)} \left( \sum_{k=1}^{m} \left( {}_k \left(p_k - x^*\right)\left(p_k - x^*\right)^{\mathrm{T}} \Sigma_k \right) - 2\sum_{k=1}^{m} \Sigma_k \right).
$$

Finally, substituting $\boldsymbol{x}^*$ with *Equation A27*, we obtain

$$
\mathsf{H} = \frac{K}{4\left(K + c(\boldsymbol{x}^*, \boldsymbol{x}^*)\right)} \left( \sum_{k=1}^{m} \left( \Sigma_k^{-1}\left(p_k - \bar{p}_w\right)\left(p_k - \bar{p}_w\right)^{\mathrm{T}} \Sigma_k^{-1} \right) - 2\sum_{k=1}^{m} \Sigma_k^{-1} \right). \tag{A42}
$$

## Special case: identically shaped Gaussian efficiency functions

For the special case that the Gaussian covariance matrices $\Sigma_k$ are equal ($\Sigma_1 = \Sigma_2 = \ldots = \Sigma_m = \Sigma_{\mathrm{G}}$), fulfilled in *Appendix 5—figure 1B*, the Hessian matrix simplifies to

$$
\begin{aligned}
\mathsf{H} &= \frac{Km}{4\left(K + c(\boldsymbol{x}^*, \boldsymbol{x}^*)\right)} \Sigma_{\mathrm{G}}^{-1} \left( \frac{1}{m}\sum_{k=1}^{m} \left( \left(p_k - \bar{p}\right)\left(p_k - \bar{p}\right)^{\mathrm{T}} \right) - 2\Sigma_G \right) \Sigma_{\mathrm{G}}^{-1} \\
&= \frac{Km}{4\left(K + c(\boldsymbol{x}^*, \boldsymbol{x}^*)\right)} \Sigma_{\mathrm{G}}^{-1} \left( \Sigma_p - 2\Sigma_G \right) \Sigma_{\mathrm{G}}^{-1},
\end{aligned} \tag{A43}
$$

where we used that $\Sigma_p = (\sum_{k=1}^{m}(p_k - \bar{p})(p_k - \bar{p})^{\mathrm{T}})/m$ is the covariance matrix of the positions $p_k$ of the pathogen vectors. Since the fraction in *Equation A43* is positive and $\Sigma_{\mathrm{G}}^{-1}$ is positive definite, it follows that the definiteness of $\mathsf{H}$ is given by the definiteness of the matrix $\Sigma_p - 2\Sigma_{\mathrm{G}}$. Hence, the singular point $\boldsymbol{x}^*$ is uninvadable whenever

$$
\Sigma_p - 2\Sigma_G < 0, \tag{A44}
$$

where the inequality indicates negative definiteness.

## Special case: maximal symmetry

For the even more special case that $\Sigma_G = \sigma_G^2 I$ and that the pathogen vectors are arranged symmetrically, resulting in $\Sigma_p = \sigma_p^2 I$, we obtain

$$H = \frac{Km}{4\sigma_G^{-4}(K + c(\boldsymbol{x}^*, \boldsymbol{x}^*))}(\sigma_p^2 - 2\sigma_G^2)I. \tag{A45}$$

In this case, the singular point $\boldsymbol{x}^*$ is a branching point whenever $\sigma_p^2 > 2\sigma_G^2$. Assuming that all $m$ pathogen vectors have an equal distance $d$ to each other implies that they are arranged in an $m - 1$-dimensional regular simplex (see *Figure 1*). For this case, $\sigma_p^2 = d^2/(2m)$, and we obtain the branching condition $d/\sigma_G > 2\sqrt{m}$. Assuming pathogen distance $d = 1$ and substitute $\sigma_G^{-1}$ as virulence $v$, we obtain

$$v > 2\sqrt{m}. \tag{A46}$$

