## [Editor Report]

This important theoretical and numerical study deals with a contemporary topic in evolutionary biology, immunology and population genetics. The structure of the models and the analytic framework used are relevant and sound, and the combination of two types of models is a powerful approach that produces compelling evidence to support the hypothesis on the role of heterozygote advantage in maintaining MHC gene polymorphism. The description of the models is easy to follow, and the paper would be of interest to specialists in evolution, immunology, and the general *eLife* readership.

---

## [Decision Letter]

**Decision letter after peer review:**

[Editors’ note: the authors submitted for reconsideration following the decision after peer review. What follows is the decision letter after the first round of review.]

Thank you for submitting your work entitled "Heterozygote advantage can explain the extraordinary diversity of immune genes" for consideration by *eLife*. Your article has been reviewed by 3 peer reviewers, one of whom is a member of our Board of Reviewing Editors, and the evaluation has been overseen by a Senior Editor. The reviewers have opted to remain anonymous.

Our decision has been reached after consultation between the reviewers. Based on these discussions and the individual reviews below, we regret to inform you that your work will not be considered further for publication in *eLife*. However, we would like to keep open the possibility for a new submission of a substantially revised manuscript.

The reviewers were somewhat divided in their assessment, but it is clear from the comments and the editorial discussion that your paper contains novel ideas and insights that could be relevant for a wide audience. The conclusion that heterozygote advantage can in principle explain high degrees of diversity at MHC loci is important. However, the reviewers also identified a number of shortcomings in the paper. Specifically:

1) The biological relevance of the assumptions underlying the model is unclear. In particular, it is not clear whether the model assumptions are realistic enough, particularly compared to previous "bitstring" approaches.

2) There are also a number of mathematical assumptions that are questionable, as pointed out in the reviews.

Overall, the scope of the model would need to be considerably expanded in order to address the points raised by the reviewers. If you choose to revise and resubmit, please address all points raised by the reviewers. We would likely send a revised version of the paper to additional reviewers for evaluation.

*Reviewer #1:*

This is a very interesting theoretical paper addressing the extraordinary amounts of polymorphism found at MHC loci. The paper shows convincingly that heterozygote advantage can generate coexistence of hundreds of different alleles when there is a tradeoff between the protection that a given allele provides against different pathogens. This result is obtained when the number of different pathogens is not small. The paper is well written, the analysis is sound, and even though the paper is purely theoretical, it will be of interest to a wide audience because of the simplicity of the theoretical principle at work, and because of the importance of the topic.

Some comments:

1. It never becomes clear what the significance is of the number of "modules" assumed to make up an allele. This number is assumed to be 10 in the paper, but it would be interesting to know how the model behaves when this number is varied. Presumably, the higher the number of modules, the higher the number of different alleles that can be maintained.

2. A number of recent papers (including ones by the senior author) have studied the effects of the dimension of phenotype space on the amount of diversity that can be maintained. The number of modules mentioned above could probably be interpreted as the dimension of phenotype space (in some sense), and based on that, I think the authors should discuss their results in the context of previous work about high-dimensional phenotype spaces.

3. The number of different pathogens, m, also plays the role of a dimension, in this case the dimension of the external environment. The authors show that as this dimension is increased, the number of alleles that can coexist also increases. Again, this finding should be discussed in the context of the effect of dimension on the amount of diversity.

4. The model assumes that the pathogens are fixed. It would be interesting to know what happens if the pathogens themselves also underwent evolutionary dynamics. I think the authors should at least speculate about the effects of including this complication in the model.

5. Another connection that should be made much clearer is the relationship of the model that the authors are using to Levene type models for the evolution of habitat specialization. There are some clear paralleles between these two modelling approaches, and this should be discussed and put into context. In fact, I think in some abstract sense, the results that the authors are reporting could be directly translated to certain ecological scenarios of habitat specialization, with the prediction that high-dimensional phenotypes and high-dimensional external habitat structure can lead to large amounts of coexisting diversity, simply based on habitat preference.

6. The authors argue (convincingly) that coexistence of different alleles is due to heterozygote advantage. In general, coexistence of the type described here is always due to some form of frequency dependence, in the sense that each of the coexisting alleles has an advantage when rare. It would be good if the authors could make this connection between heterozygote advantage and frequency dependence explicit (with the latter being a more general concept).

7. The authors mention that they assume multiplicativity for the effects of pathogens (Equation1). Does the whole theory break down with additivity? It would be good to include more comments on this.

8. Likewise, the authors assume codominance in Equation 1. What about other dominance modes? E.g. what about multiplicative effects of the two alleles? Presumably, the results would be rather different in this case…

9. l. 98: the indices i and j appear out of the blue at this point, and it is not clear in what sense they characterize an individual.

*Reviewer #2:*

Review of Heterozygote advantage can explain the extraordinary diversity of immune genes.

First I should say that I focus on the immunological aspects of this paper and am not familiar with the population genetic aspects and leave these to a referee more versed in the area. There are also a number of theoretical immunologists who have worked in the area of immune recognition by MHC who would likely know this area better than myself. In particular I would refer to Borgans and De Boer's work in this area.

I found it hard to follow the biological underpinnings of some of the paper and will try to explain why. My questions are indicated on lines beginning with a ?

The paper tackles the problem of understanding the causes of MHC diversity. Why are there so many alleles at each locus of the MHC. It would be good if the authors gave us an idea of the extent of the number of alleles that are found at each MHC locus, so we can get an idea of the magnitude of the diversity we are trying to explain. The approach, as far as I could tell is as follows:

1. The authors use what looks similar to previous bitstring model to describe the extent to which a given MHC provides protection against a given pathogen. Each MHC is given by an m-dimensional vector m as is each pathogen which is given by p.

2. The efficiency of a given MHC in controlling a given pathogen decreases with increasing euclidian distance between the vectors m and p. I.e. e(p-x)

3. The overall efficiency of a given individual against a pathogen is the sum of the efficiencies of its two MHC alleles at controlling the pathogen.

4. The efficiency of an individual against an ensemble of m pathogen c is the product of the efficiency of that individual for controlling each pathogen.

This is basically equation (i)

5. Selection is a function of c above.

6. They follow the evolutionary dynamics of alleles of MHC in a diploid population of size N.

7. The key results are in Figure 3. The degree of MHC diversity maintained increases with increases in the number of pathogens the degree of pathogen dissimilarity and occurs at intermediate levels of the half saturation constant K.

Comments in order of being encountered in the paper.

1. Why not use the earlier bitstring models rather than a new model. What is the justification of letting the elements of m and p be real numbers (and what determines how large these numbers can be?).

2. Why have c_max or whether the 2 is needed in Equation 1

3. What is the degree of selection that the pathogens chosen impose? Is it biologically reasonable given what we know about MHC?

4. The number of pathogens appears limited to a max of 8 why? How does it scale for more reasonable numbers.

5. Why have models with symmetry in distances between pathogens as in version 1.

6. I don't see the need for recombination … there can be different MHC generated by mutation.

7. Is it necessary to use an idea of generalist – it is supported by a single paper and not potentially yet part of mainstream immunology.

8. From 3 it seems that a key feature is the degree of dissimilarity. A search of the document does not find it being defined? What are biologically plausible values for this parameter?

9. Does the model make any predictions that allow discrimination or rejection of current hypotheses?

Overall I found it hard to follow the paper. To a large extent may be because I am not an expert in population genetics. More problematic was the difficulty I found understanding the details of the connection with immunology and lack of parameterization of biologically reasonable parameter regimes. I would be more convinced about the claims of the authors if the above questions were addressed. Finally I believe that either Jose Borghans or Rob De Boer would be very well suited to review the paper.

*Reviewer #3:*

There are good reasons to assume that individuals with two different alleles at an MHC locus are better protected against pathogen-induced disease (and, hence, have a higher fitness) than individuals that are homozygous for any of these alleles. Based on this, it has been argued that such heterozygote advantage may explain the high degree of polymorphism found at MHC loci. However, there is a problem with this argument. In principle, an arbitrary large number of alleles can be kept in the population by heterozygote advantage, but this is only possible if (1) all heterozygotes have a similarly high fitness, while (2) all homozygotes have a similarly low fitness. If the fitness effects of MHC alleles are drawn at random, it is unlikely that these conditions are satisfied. For this reason, earlier studies concluded that heterozygote advantage is most likely not the only factor explaining the high degree of MHC diversity. In the present study, the authors demonstrate that, in principle, the heterozygote-advantage hypothesis can be rescued if the current MHC alleles did not 'fall from the air' (i.e. were created by random mutation of large effect size) but instead were shaped by gradual evolution (based on the rare influx of mutations of small effect size). The authors argue that (under suitable conditions) this evolution process has the tendency to shape MHC alleles in such a way that conditions (1) and (2) are satisfied, hence allowing the stable coexistence of many (even hundreds) of MHC alleles.

The study shows that probability calculations, as in earlier studies, have to be taken with care, since the current MHC alleles may not be a random sample from a universe of possibilities but rather the product of diversifying evolution. This insight is interesting and important.

I am less impressed by the model that is used to illustrate their point. Most importantly, fitness (survival) is assumed to be a function of 'condition,' which in turn is defined in a rather specific manner (Equations 1 and 2). To me, it would have been plausible to define survival directly by the product term in Equation 1 (overall survival results from surviving many pathogen-induced challenges), but now this term is plugged into a saturating Michaelis-Menten function. Due to this transformation (which is not motivated well), conditions (1) and (2) are much more easily satisfied. I would like to see the same analysis, but now for the more plausible assumption that condition corresponds to viability. I would also have appreciated if the authors had applied their method to other fitness functions, like the one in De Boer et al. (2004), in order to work out more clearly why the authors arrive at conclusions that contrast with those of earlier studies.

The authors use an adaptive dynamics approach for modelling the fine-tuning of MHC alleles. I wonder whether this kind of approach, which is based on continuously varying traits and very rare mutations of very small effect size is realistic in the context of MHC evolution. Mutations of MHC alleles seem to occur frequently, and even point mutations do often have a large effect. In other words, the genotypic-phenotype mapping is most probably quite intricate, making some baseline assumptions of the model (e.g. Gaussian distribution of effect sizes) questionable. The study would have been more convincing if the authors would have incorporated more realistic assumptions on the action of MHC alleles in their individual-based simulations, rather than just mimicking the assumptions of their analytical model.

[Editors’ note: further revisions were suggested prior to acceptance, as described below.]

Thank you for resubmitting your work entitled "Heterozygote advantage can explain the extraordinary diversity of immune genes" for further consideration by *eLife*. Your revised article has been evaluated by Detlef Weigel (Senior Editor) and a Reviewing Editor.

The manuscript has been improved but there are some remaining issues that need to be addressed, as outlined below:

*Reviewer #2 (Recommendations for the authors):*

The MHC genes encode proteins that bind antigenic oligopeptides and present them to receptors on immune cells. Successful presentation of antigens to immune T cells by the MHC is the key stage in the immune response to pathogens. While T cell receptors are variable and can adapt during the process of clonal selection, the MHC repertoire is inherited from parents and does not change during an individual's lifetime. Any pathogen genotype that escapes presentation by the MHC is able to reproduce at a higher rate due to the lack of initiation of an effective immune response. MHC genes are under strong natural selection for efficient binding of pathogen antigens, but surprisingly are often represented by hundreds of alleles (variants) in natural populations of vertebrates. In addition, MHC genes are often characterised by extremely long genealogies. These two features of MHC genes are particularly difficult to reconcile with strong natural selection, which usually results in low genetic polymorphism and often a short coalescence time. The paper by Sijestam and Rueffler investigates the role of one of the key balancing selection forces, the so-called heterozygote advantage, which has been hypothesised to maintain MHC gene polymorphism.

Strengths

The manuscript is clearly written and deals with a current and important topic in evolutionary biology, immunology and population genetics.

The structure of the models presented and the framework used are sound, with one of the models based on a classic theoretical work in the field and predict results that contradict the majority of recent theory on the evolution of MHC gene polymorphism.

The description of the models is easy to follow. The combination of two types of models ('Gaussian model' and 'bit-string model') is a powerful approach to test the hypothesis on the role of heterozygote advantage in maintaining MHC gene polymorphism.

The authors list two key assumptions of their model, which they believe are responsible for the conclusion of the paper. However, only one of these assumptions actually provides fundamental support for the role of heterozygote advantage in maintaining MHC gene polymorphism. Recent models show that the Red Queen process is able to maintain high levels of MHC polymorphism despite the mortality of hosts that respond poorly to infections (see detailed comments).

The work has the potential to shed new light on the long-standing debate about the importance of the Red Queen process vs. heterozygote advantage as a balancing selective force that maintains MHC gene polymorphism, although further analysis is needed to address concerns about some of the model assumptions made.

Weaknesses

There are two fundamental weaknesses in the manuscript, as the models in their current form and without additional analyses do not have sufficient scientific strength to restore the heterozygote advantage as a force capable of maintaining high genetic polymorphism of the MHC. First, because some of the assumptions used to model heterozygous advantage differ from those used in previous studies (suggestions I, II below). Second, because there is no reference to the Red Queen, which a balancing selection force currently receiving considerable attention in studies of MHC polymorphism (suggestion III). I also strongly believe that my concerns about the current limited power of the conclusions presented by the authors can be addressed by adding the suggested analyses. This could be quite a lot of work, but the analyses do not need to be performed in the full parameter space but are mandatory for the paper to be of the highest scientific quality. Here is a short list of the analyses that need to be added (see below for a more detailed description): (I.) In the Gaussian model, add analysis of pathogen peptides evolving by genetic drift by allowing Brownian motion to move the pathogen distribution in allele trait space. (II.) In the bit-string model, add analysis of pathogens evolving by genetic drift. (III.) Add simulations with only the Red Queen process operating (e.g. by simulating haploid hosts co-evolving with pathogens). However, it is necessary that all current assumptions of the bit-string model remain unchanged (including the relationship between infection outcome and host fitness), except for the addition of pathogens co-evolving with hosts.

- The authors do not communicate, or perhaps do so in a way that can be easily overlooked by the reader, the fundamental conceptual challenge associated with testing the role of heterozygote advantage in host-pathogen evolution. By this, I mean that heterozygote advantage can only be modelled or tested in the absence of host-pathogen coevolution, because coevolution generates negative frequency-dependent selection (the Red Queen process). Without this explanation, the fact that pathogens do not evolve in the model may seem ridiculous to the reader. This aspect needs to be clear from the point at which the model is described, or perhaps even from the introduction.

- In this model, pathogens do not reproduce, and I have explained above why authors cannot indeed link pathogen fitness to the host immune response if they want to test heterozygote advantage in the absence of the Red Queen process. However, as I understood from the description, pathogens in the bit-string model consist of random pools of peptides: "In the bit-string model, the m pathogens are each given npep randomly drawn bit-strings", and in the Gaussian model, the centres of the pathogens do not change their position in the allele trait space. The consequence of this assumption would be an evolutionary process that stops far in the future (pathogens are constant over time). Even if I am wrong and pathogens are randomly generated in each generation, this will still be incorrect because the model artificially assumes the maximum possible genetic diversity of pathogen peptides, which combined with the way fitness is calculated (similar to a geometric mean) would only favour a large number of alleles under heterozygote advantage. Furthermore, such an unrealistically high genetic diversity of oligopeptides simulates pathogen populations of much larger effective size than in real populations. A proper model for modelling heterozygote advantage would require modelling pathogens evolving by genetic drift. Even if the authors do not agree with my point, they should include the scenario in which pathogens evolve by genetic drift, as this has been assumed in previous models addressing the question of the role of heterozygote advantage and Red Queen on MHC polymorphism. Pathogens evolving by genetic drift are easily implemented in the bit-string model. In the Gaussian model, I would suggest introducing Brownian motion of pathogen distributions. It is very unrealistic to assume that pathogens would occupy a stationary point in the allelic trait space over so many generations, rather than moving.

- Host fitness is calculated using geometric mean mechanics, which forces the MHC allele (in a hypothetic haploid individual) to be located in the middle of the pathogen distributions located in allele trait space. This is a very strong assumption and differs from other theoretical studies of MHC gene polymorphism (e.g. Borghans et al. 2004, Ejsmond and Radwan 2015). Note, that I do not say that this is an incorrect assumption. Perhaps natural systems differ in the way infections affect host fitness. Anyway, the lack of modelling of other balancing selection forces in the presented work, namely the Red Queen (negative frequency dependence), leaves the reader with a big question mark as to the extent to which the result presented by authors is driven by the specificity of the model or the assumption that immune response and fitness scale in a 'geometric mean'-like manner. Thus, I think the Red Queen process scenario needs to be added, and my suggestion should not be taken as a suggestion of adding an additional analysis beyond the scope of this paper. Each model has its own specificity and it is necessary to know what the levels of polymorphism are when the Red Queen process is simulated. My expectation would be that if fitness is calculated in the way the authors did, the Red Queen process will maintain few alleles. This would be a strong message emerging from this study and the previous literature that heterozygote advantage or the Red Queen process can maintain polymorphism depending on the distribution of fitness effects of infections by different pathogen species.

- The authors should comment on the fact that the MHC gene polymorphism in their model would break down in the presence of several highly virulent pathogen species (narrow distributions in the Gaussian model) and under more realistic assumptions for pathogen classes e.g. helminths, i.e. a large number of antigenic oligopeptides (Figure 5 for v=5).

- The assumption (a) that pathogens are lethal in the absence of an appropriate immune response does not support the role of heterozygote advantage in maintaining MHC gene polymorphism as suggested by the authors. See the model of Ejsmond et al. 2023 ('Adaptive immune response selects for postponed maturation and increased body size'), where hosts with a poor response to pathogens also die, but high MHC gene polymorphism is maintained in particular by the Red Queen process. However, note also that Ejsmond et al. assumed that infections affect fitness proportionally, which is contrary to assumption (b) in the reviewed paper. I would suggest reducing the importance of the first assumption or discussing alternative models that show that the Red Queen process is able to maintain high MHC gene polymorphism despite assumption (a). I believe that the difference in assumption (b) is fundamental to the conclusions derived from this paper and other models.

- Show information about the proportion of hosts in the population dying per generation under main simulated scenarios

- Consider renaming 'allelic trait space' to epitope or agretope space

---

## [Author Response]

[Editors’ note: the authors resubmitted a revised version of the paper for consideration. What follows is the authors’ response to the first round of review.]

The reviewers were somewhat divided in their assessment, but it is clear from the comments and the editorial discussion that your paper contains novel ideas and insights that could be relevant for a wide audience. The conclusion that heterozygote advantage can in principle explain high degrees of diversity at MHC loci is important. However, the reviewers also identified a number of shortcomings in the paper. Specifically:1) The biological relevance of the assumptions underlying the model is unclear. In particular, it is not clear whether the model assumptions are realistic enough, particularly compared to previous "bitstring" approaches.2) There are also a number of mathematical assumptions that are questionable, as pointed out in the reviews.Overall, the scope of the model would need to be considerably expanded in order to address the points raised by the reviewers. If you choose to revise and resubmit, please address all points raised by the reviewers. We would likely send a revised version of the paper to additional reviewers for evaluation.

We appreciate the opportunity to revise our manuscript and thank the reviewers for their insightful comments. We have made significant revisions to the manuscript, and are confident that we have adequately addressed the reviewers' comments (see especially our responses to Reviewer 2 and Reviewer 3 below).

First, we have clarified the ecological assumptions underlying our model and provided stronger motivation and interpretation from a biological standpoint. Specifically, we highlight two crucial assumptions about how pathogens affect the host, which form the basis for our results:

Each pathogen is assumed to be able to cause mortality if not kept at bay by the immune system.The combined effect of multiple pathogens on host survival exceeds the sum of the effects of each pathogen alone.

We refer to our response to reviewer 3 for more a more detailed motivation. Under assumption(a) and (b), we show that fitness values emerge such that heterozygote advantage (HA) alone can lead to very high allelic diversity. These assumptions are, in our model, realized by Equation4 and 5, respectively.

Second, we now include a completely new section where we present a ‘bitstring’ type model to more mechanistically model MHC interactions with pathogens. This bitstring-approach has been used in several prior studies (e.g., Borghans *et al.* 2004, Ejsmond and Radwan 2011, 2015). Our 'bitstring' model qualitatively reproduces the results of our original Gaussian model, thereby confirming that the observed high levels of allelic polymorphism are not due to some specific dynamics exclusive to our Gaussian model, but are a more general outcome resulting from fitness assumptions (a) and (b).

Reviewer #1 (General assessment and major comments (Required)):This is a very interesting theoretical paper addressing the extraordinary amounts of polymorphism found at MHC loci. The paper shows convincingly that heterozygote advantage can generate coexistence of hundreds of different alleles when there is a tradeoff between the protection that a given allele provides against different pathogens. This result is obtained when the number of different pathogens is not small. The paper is well written, the analysis is sound, and even though the paper is purely theoretical, it will be of interest to a wide audience because of the simplicity of the theoretical principle at work, and because of the importance of the topic.

Thank you very much for the positive evaluation of our work.

Some comments:1. It never becomes clear what the significance is of the number of "modules" assumed to make up an allele. This number is assumed to be 10 in the paper, but it would be interesting to know how the model behaves when this number is varied. Presumably, the higher the number of modules, the higher the number of different alleles that can be maintained.

We agree that these parameters can be interpreted as dimensions in the phenotype and environment spaces, respectively.

In the revised version, we do not use the term “module” anymore and we reformulated our explanation for why we consider trait spaces that have the dimension one less than the number of pathogens.

2. A number of recent papers (including ones by the senior author) have studied the effects of the dimension of phenotype space on the amount of diversity that can be maintained. The number of modules mentioned above could probably be interpreted as the dimension of phenotype space (in some sense), and based on that, I think the authors should discuss their results in the context of previous work about high-dimensional phenotype spaces.3. The number of different pathogens, m, also plays the role of a dimension, in this case the dimension of the external environment. The authors show that as this dimension is increased, the number of alleles that can coexist also increases. Again, this finding should be discussed in the context of the effect of dimension on the amount of diversity.

We believe that the reviewer is referring, amongst others, to the following papers:

Doebeli and Ispolatov (2010 Science)

Debarre et al. (2014 AmNat)

Svardal et al. (2014 Evolution)

These papers investigate in a resource competition context whether the dimensionality of the consumer trait space facilitates whether a monomorphic consumer population splits into two, but do not address how the dimensionality of the trait space affects final diversity. Furthermore, in these studies, evolutionary branching is driven by negative frequency-dependent selection at the level of the phenotypes. This is not the case in our model, where branching is driven by HA. We therefore chose to not discuss our results in the context of the papers mentioned above.

Débarre F, Nuismer SL, Doebeli M. Multidimensional (co)evolutionary stability. Am Nat. 2014 Aug;184(2):158-71. doi: 10.1086/677137. Epub 2014 Jul 8. PMID: 25058277.

Doebeli M, Ispolatov I. Complexity and diversity. Science. 2010 Apr 23;328(5977):494-7. doi:

10.1126/science.1187468. PMID: 20413499.

Svardal H, Rueffler C, Doebeli M. Organismal complexity and the potential for evolutionary diversification. Evolution. 2014 Nov;68(11):3248-59. doi: 10.1111/evo.12492. Epub 2014 Sep 2. PMID: 25087681.

4. The model assumes that the pathogens are fixed. It would be interesting to know what happens if the pathogens themselves also underwent evolutionary dynamics. I think the authors should at least speculate about the effects of including this complication in the model.

In the discussion of the revised manuscript, we have included a paragraph discussing evolutionary dynamics in the pathogens [Line 450-465]. We speculate that evolving pathogens could lead to regions of the parameter space where polymorphism can evolve by HA.

5. Another connection that should be made much clearer is the relationship of the model that the authors are using to Levene type models for the evolution of habitat specialization. There are some clear paralleles between these two modelling approaches, and this should be discussed and put into context. In fact, I think in some abstract sense, the results that the authors are reporting could be directly translated to certain ecological scenarios of habitat specialization, with the prediction that high-dimensional phenotypes and high-dimensional external habitat structure can lead to large amounts of coexisting diversity, simply based on habitat preference.

At a first glance, our model may seem similar to Levene-type island models. However, we disagree that there is a structural similarity to our model. Polymorphism in island models is generally driven by negative frequency dependence acting at the level of the phenotype, resulting in local adaptation. In contrast, polymorphism in our model is driven by HA arising from selection to resist all pathogens. As a consequence, in our model fitness is strictly multiplicative while in island models fitness has a strong additive component.

6. The authors argue (convincingly) that coexistence of different alleles is due to heterozygote advantage. In general, coexistence of the type described here is always due to some form of frequency dependence, in the sense that each of the coexisting alleles has an advantage when rare. It would be good if the authors could make this connection between heterozygote advantage and frequency dependence explicit (with the latter being a more general concept).

There are indeed different forms of negative frequency dependence leading to coexistence, where HA is just one. We now discuss the connection between HA and the more general concept of negative frequency dependence in more detail [Line 426-449].

7. The authors mention that they assume multiplicativity for the effects of pathogens (Equation1). Does the whole theory break down with additivity? It would be good to include more comments on this.

We now state explicitly that our results rely on assumptions (a) and (b) (as stated in our response to the editor). Both of these assumptions are essential for the coexistence of a large number of alleles. The first assumption (a) posits that a single pathogen can be lethal if an immune response is lacking. In our model, this assumption is fulfilled due to the multiplicativity in Equation (4). Indeed, assuming that pathogens affect host fitness additively dramatically reduce the number of coexisting alleles.

8. Likewise, the authors assume codominance in Equation 1. What about other dominance modes? E.g. what about multiplicative effects of the two alleles? Presumably, the results would be rather different in this case…

We choose codominance in our model because it seems to be a valid representation for MHC alleles. Dominance results in a higher degree of HA compared to codominance, which in turn leads to a slightly larger number of coexisting alleles (results not shown). In our study, we chose codominance as it provides more conservative estimates regarding the extent of allelic polymorphism. This rationale is now explicitly addressed in the revised manuscript at Line 161-165.

To the best of our knowledge, there is no literature suggesting a multiplicative effect between two alleles at an MHC locus. In our model, efficiencies range from 0 to 1. Utilizing multiplicative effects, rather than codominance, would result in a model akin to recessivity but even more restrictive. The prevailing theories regarding MHC alleles, such as the HA hypothesis (Doherty and Zinkernagel 1975) and the rare allele advantage hypothesis (Apanius *et al.* 2017), suggest that an individual can benefit already from a single allele at a diploid MHC locus.

To offer a simplified mechanistic explanation: each allele at an MHC locus produces its own specific MHC molecule, regardless of whether it is present in one or two copies. This leaves us with two scenarios: either the amount of gene product is proportional to the number of alleles or, due to dosage compensation, a single copy or two copies result in the same amount of gene product. These two cases correspond to codominance or full dominance, respectively.

Apanius V, Penn D, Slev PR, Ruff LR, Potts WK. The nature of selection on the major histocompatibility complex. Crit Rev Immunol. 1997. 17:179-224.

Doherty PC, Zinkernagel RM. Enhanced immunological surveillance in mice heterozygous at the H-2 gene complex. Nature. 1975. 256:50-2.

9. l. 98: the indices i and j appear out of the blue at this point, and it is not clear in what sense they characterize an individual.

We have now clarified the meaning of i and j, which are giving the index of the alleles of an individual [L. 169]

Reviewer #2 (General assessment and major comments (Required)):Review of Heterozygote advantage can explain the extraordinary diversity of immune genes.First I should say that I focus on the immunological aspects of this paper and am not familiar with the population genetic aspects and leave these to a referee more versed in the area. There are also a number of theoretical immunologists who have worked in the area of immune recognition by MHC who would likely know this area better than myself. In particular I would refer to Borgans and De Boer's work in this area.I found it hard to follow the biological underpinnings of some of the paper and will try to explain why. My questions are indicated on lines beginning with a ?The paper tackles the problem of understanding the causes of MHC diversity. Why are there so many alleles at each locus of the MHC. It would be good if the authors gave us an idea of the extent of the number of alleles that are found at each MHC locus, so we can get an idea of the magnitude of the diversity we are trying to explain. The approach, as far as I could tell is as follows:1. The authors use what looks similar to previous bitstring model to describe the extent to which a given MHC provides protection against a given pathogen. Each MHC is given by an m-dimensional vector m as is each pathogen which is given by p.2. The efficiency of a given MHC in controlling a given pathogen decreases with increasing euclidian distance between the vectors m and p. I.e. e(p-x)3. The overall efficiency of a given individual against a pathogen is the sum of the efficiencies of its two MHC alleles at controlling the pathogen.4. The efficiency of an individual against an ensemble of m pathogen c is the product of the efficiency of that individual for controlling each pathogen.This is basically equation (i)5. Selection is a function of c above.6. They follow the evolutionary dynamics of alleles of MHC in a diploid population of size N.7. The key results are in Figure 3. The degree of MHC diversity maintained increases with increases in the number of pathogens the degree of pathogen dissimilarity and occurs at intermediate levels of the half saturation constant K.Comments in order of being encountered in the paper.1. Why not use the earlier bitstring models rather than a new model. What is the justification of letting the elements of m and p be real numbers (and what determines how large these numbers can be?).

In our revised manuscript, we have included a completely new section analysing a bitstring model, inspired by Borghans et al. (2004). The bitstring model has both pros and cons and serves as a useful complement to our Gaussian model. The bit-string approach offers a more mechanistic way to define the trade-offs involved in inducing an immune defence against different pathogens. It does, however, not allow for an analytical analysis and also leads to highly specific behaviour in the trade-off dynamics. In contrast, our Gaussian model is not mechanistic but offers a more mathematically flexible definition of the trade-offs, making fewer assumptions in the process.

In the revised manuscript, we demonstrate that the bit-string model yields qualitatively similar results to the Gaussian model, showing that the key to our findings is the combination of host-pathogen assumptions (a) and (b), (see above), as well as the presence of some form of trade-off, whether modelled through the Gaussian or bit-string approach.

2. Why have c_max or whether the 2 is needed in Equation 1

In our model, *c*_max_ serves as a scaling constant to standardize the condition of the optimal homozygote *c*(*x**, *x**) to 1.

For our purpose, we are interested in situations where – during the evolutionary process – realized survival probabilities cover a large range of the interval between zero and one. One way to achieve this, is, since evolution passes through the generalist allele, to standardize the condition of individuals homozygous for the generalist allele such that it is in the vicinity of *K*, the condition value for which survival equals one half. By choosing *c*_max_ such that *c(x*, x*)=1*, we can now choose *K*-values such that our goal is fulfilled, as visible from the x-axes in the lower panel of Figure 3 and in Figure 6B-D.

The division by 2 in Equation4 is included to reflect the assumption of codominance, whereby the average phenotype is expressed when two different alleles are present

3. What is the degree of selection that the pathogens chosen impose? Is it biologically reasonable given what we know about MHC?

In our model, the degree of selection imposed by pathogens does not depend on any fixed values, but is an emergent property. If our host-pathogen assumptions (a) and (b) are fulfilled and the trade-off in a host’s ability to adapt to different pathogens is sufficiently strong, then evolution results in allelic values that are able to coexist.

4. The number of pathogens appears limited to a max of 8 why? How does it scale for more reasonable numbers.

In our Gaussian model, we limit the number of pathogens to eight because a high level of allelic coexistence is already observed at this point. In this version of our model, pathogens are clearly distinct, representing different taxa rather than merely different species or strains. Therefore, a larger number of pathogens is not essential for the Gaussian model to generate allelic diversity.

In contrast, the bit-string model allows for more overlap between pathogens, making them better suited to represent closely related species or even different strains of the same species. In this model, we go up to 100 pathogens, observing similar results to those in the Gaussian model but requiring a higher number of pathogens for high MHC diversity to evolve. We now elaborated upon these points in the revised manuscript at Line 352-359.

5. Why have models with symmetry in distances between pathogens as in version 1.

Our qualitative results are unaffected by assuming symmetric or asymmetric pathogen optima. Given this, we opted to present the symmetric case in the main text, and include the results for different non-symmetric scenarios in the supplementary material. This is explained at Line 140-143 in the revised manuscript.

6. I don't see the need for recombination … there can be different MHC generated by mutation.

We now exclude recombination from the model, as it does indeed not provide anything qualitatively different from point mutations.

7. Is it necessary to use an idea of generalist – it is supported by a single paper and not potentially yet part of mainstream immunology.

In the initial Gaussian model, we naturally get the possibility for a generalist allele that is intermediately placed between the different pathogen optima due to the continues nature of the trait space. In our newly introduced bit-string model, which operates in a trait space of a non-continues nature, generalists alleles do not generally exist, particularly in parameter regions where trade-offs exist (warm-coloured regions in Figure 5). Despite this, the bit-string model still replicates the results of high allelic diversity. Therefore, the notion of a generalist is more of an emergent property of the Gaussian model, and although it has some empirical support (Chappell *et al.* 2015, *eLife* 4:e05345), it is not necessary for generating high MHC diversity in our study.

8. From 3 it seems that a key feature is the degree of dissimilarity. A search of the document does not find it being defined? What are biologically plausible values for this parameter?

In the revised manuscript, we have altered the model presentation. In the original manuscript, the dissimilarity *d* was defined as the distance between the Gaussian optima. In the revised version, we achieve the same effect by varying the slope of the Gaussian functions (*v* = *σ*
^-1^), while keeping the distance fixed. This is mathematically equivalent to the original approach. We now refer to this new parameter *v* as pathogen virulence. All these details, and the biological interpretation of virulence *v* are described and defined in Line 129-139 and the legend of Figure 1.

9. Does the model make any predictions that allow discrimination or rejection of current hypotheses?

Our primary aim is to investigate whether heterozygote advantage (HA) can, in principle, maintain the high degrees of MHC polymorphism observed in nature. The prevailing hypothesis suggests that HA alone cannot achieve this. Our model challenges this view, suggesting that HA can indeed maintain such high levels of diversity, provided that assumptions (a) and (b) are met, and that a trade-off in immune performance exists. We therefore reject the prevailing hypothesis.

Overall I found it hard to follow the paper. To a large extent may be because I am not an expert in population genetics. More problematic was the difficulty I found understanding the details of the connection with immunology and lack of parameterization of biologically reasonable parameter regimes. I would be more convinced about the claims of the authors if the above questions were addressed. Finally I believe that either Jose Borghans or Rob De Boer would be very well suited to review the paper.

We are thankful that the reviewer took the time to read and evaluate our manuscript from the perspective of their expertise. By addressing their questions, we found many opportunities to improve the quality of our manuscript.

Reviewer #3 (General assessment and major comments (Required)):There are good reasons to assume that individuals with two different alleles at an MHC locus are better protected against pathogen-induced disease (and, hence, have a higher fitness) than individuals that are homozygous for any of these alleles. Based on this, it has been argued that such heterozygote advantage may explain the high degree of polymorphism found at MHC loci. However, there is a problem with this argument. In principle, an arbitrary large number of alleles can be kept in the population by heterozygote advantage, but this is only possible if (1) all heterozygotes have a similarly high fitness, while (2) all homozygotes have a similarly low fitness. If the fitness effects of MHC alleles are drawn at random, it is unlikely that these conditions are satisfied. For this reason, earlier studies concluded that heterozygote advantage is most likely not the only factor explaining the high degree of MHC diversity. In the present study, the authors demonstrate that, in principle, the heterozygote-advantage hypothesis can be rescued if the current MHC alleles did not 'fall from the air' (i.e. were created by random mutation of large effect size) but instead were shaped by gradual evolution (based on the rare influx of mutations of small effect size). The authors argue that (under suitable conditions) this evolution process has the tendency to shape MHC alleles in such a way that conditions (1) and (2) are satisfied, hence allowing the stable coexistence of many (even hundreds) of MHC alleles.The study shows that probability calculations, as in earlier studies, have to be taken with care, since the current MHC alleles may not be a random sample from a universe of possibilities but rather the product of diversifying evolution. This insight is interesting and important.

Thank you very much for this succinct summary of our study and your positive evaluation.

I am less impressed by the model that is used to illustrate their point. Most importantly, fitness (survival) is assumed to be a function of 'condition,' which in turn is defined in a rather specific manner (Equations 1 and 2). To me, it would have been plausible to define survival directly by the product term in Equation 1 (overall survival results from surviving many pathogen-induced challenges), but now this term is plugged into a saturating Michaelis-Menten function. Due to this transformation (which is not motivated well), conditions (1) and (2) are much more easily satisfied.

We acknowledge that the previous version of our manuscript did not provide adequate motivation for some of our biological assumptions. To improve this, we have substantially revised the manuscript and clarified how pathogens affect host fitness:

Pathogens are assumed to be lethal in the absence of an appropriate immune defence.The combined effect of multiple pathogens on host survival exceeds the sum of the effects of each pathogen alone. In other words, the effects of pathogens are not independent of each other.

Assumption results from our Equation (5) , which is criticised by the reviewer. Instead, the reviewer suggested equating the product of the immune efficiencies (Equation 4) directly with survival. Under this assumption, each factor in Equation (4) describes the probability to survive an infection by the corresponding pathogen. While this may seem more straightforward, we argue that it could in fact be less realistic. Equating survival with the product of the probabilities to survive each pathogen implies that the effects of multiple pathogens on a host are strictly independent. Under this formulation, cleared infections have no effect on surviving future infections. While this might apply for some diseases, we argue that it does not apply in general. Instead, we consider the case that a host clearing a pathogen is in a weaker condition afterwards. For example, hosts that have cleared a parasite infection might have lost weight in this process, which affects their ability to cope with future infections. It is for this reason that we model survival as a saturating function of condition. Indeed, under Equation (5), decreasing condition leads to increasingly larger effects on survival. Hence, we believe there is good reason to expect situations in nature where assumption (b) holds true.

I would like to see the same analysis, but now for the more plausible assumption that condition corresponds to viability. I would also have appreciated if the authors had applied their method to other fitness functions, like the one in De Boer et al. (2004), in order to work out more clearly why the authors arrive at conclusions that contrast with those of earlier studies.

In our revised manuscript, we incorporated this suggestion (which was also made by reviewer 2) and included a new section featuring a ‘bit-string’ model using a fitness function inspired by Borghans et al. (2004) and discussed in De Boer et al. (2004). Our 'bit-string' model reproduces the results of our original Gaussian model, showing that high levels of allelic polymorphism are not specific to how the immune efficiencies are modelled but are a more general outcome resulting from assumptions (a) and (b).

To show more directly that our assumptions (a) and (b) are necessary for high allelic diversity to evolve under heterozygote advantage (HA), we include a comparison of our bit-string model with that of Borghans et al. (2004) (Supplementary Information 5). Our assumption about how matching of bit-strings between hosts and pathogens affect pathogen recognition are identical to Borghans et al. (2004) but in their model host survival does not fulfil our assumptions (a) and (b). While Borghans et al. (2004) report up to seven alleles we find, using the same parameters, well over 100 coexisting alleles.

In conclusion, heterozygote advantage can maintain high levels of allelic diversity, provided assumptions (a) and (b) are meet. Importantly, we believe that these assumptions can be well motivated on biological grounds.

The authors use an adaptive dynamics approach for modelling the fine-tuning of MHC alleles. I wonder whether this kind of approach, which is based on continuously varying traits and very rare mutations of very small effect size is realistic in the context of MHC evolution. Mutations of MHC alleles seem to occur frequently, and even point mutations do often have a large effect. In other words, the genotypic-phenotype mapping is most probably quite intricate, making some baseline assumptions of the model (e.g. Gaussian distribution of effect sizes) questionable. The study would have been more convincing if the authors would have incorporated more realistic assumptions on the action of MHC alleles in their individual-based simulations, rather than just mimicking the assumptions of their analytical model.

We would like to point out that we already demonstrated that high MHC diversity can evolve even with relatively large mutational steps (see Figure S2). This was based on our Gaussian model, which does use continuous traits.

Furthermore, in the bit-string model included in our revision the genotype-phenotype map is indeed “intricate” and non-continuous. Specifically, single mutations can have large effects on immune defence. In the bit-string model, MHC binding efficiency is represented mechanistically, and even the smallest mutational change—altering just one bit of the bit-string—can have a substantial impact on the MHC molecule's ability to bind to different pathogen peptides. Importantly, we find that our main results hold true even under these more intricate conditions.

[Editors’ note: what follows is the authors’ response to the second round of review.]

The manuscript has been improved but there are some remaining issues that need to be addressed, as outlined below:Reviewer #2 (Recommendations for the authors):The MHC genes encode proteins that bind antigenic oligopeptides and present them to receptors on immune cells. Successful presentation of antigens to immune T cells by the MHC is the key stage in the immune response to pathogens. While T cell receptors are variable and can adapt during the process of clonal selection, the MHC repertoire is inherited from parents and does not change during an individual's lifetime. Any pathogen genotype that escapes presentation by the MHC is able to reproduce at a higher rate due to the lack of initiation of an effective immune response. MHC genes are under strong natural selection for efficient binding of pathogen antigens, but surprisingly are often represented by hundreds of alleles (variants) in natural populations of vertebrates. In addition, MHC genes are often characterised by extremely long genealogies. These two features of MHC genes are particularly difficult to reconcile with strong natural selection, which usually results in low genetic polymorphism and often a short coalescence time. The paper by Sijestam and Rueffler investigates the role of one of the key balancing selection forces, the so-called heterozygote advantage, which has been hypothesised to maintain MHC gene polymorphism.StrengthsThe manuscript is clearly written and deals with a current and important topic in evolutionary biology, immunology and population genetics.The structure of the models presented and the framework used are sound, with one of the models based on a classic theoretical work in the field and predict results that contradict the majority of recent theory on the evolution of MHC gene polymorphism.The description of the models is easy to follow. The combination of two types of models ('Gaussian model' and 'bit-string model') is a powerful approach to test the hypothesis on the role of heterozygote advantage in maintaining MHC gene polymorphism.The authors list two key assumptions of their model, which they believe are responsible for the conclusion of the paper. However, only one of these assumptions actually provides fundamental support for the role of heterozygote advantage in maintaining MHC gene polymorphism. Recent models show that the Red Queen process is able to maintain high levels of MHC polymorphism despite the mortality of hosts that respond poorly to infections (see detailed comments).The work has the potential to shed new light on the long-standing debate about the importance of the Red Queen process vs. heterozygote advantage as a balancing selective force that maintains MHC gene polymorphism, although further analysis is needed to address concerns about some of the model assumptions made.WeaknessesThere are two fundamental weaknesses in the manuscript, as the models in their current form and without additional analyses do not have sufficient scientific strength to restore the heterozygote advantage as a force capable of maintaining high genetic polymorphism of the MHC. First, because some of the assumptions used to model heterozygous advantage differ from those used in previous studies (suggestions I, II below). Second, because there is no reference to the Red Queen, which a balancing selection force currently receiving considerable attention in studies of MHC polymorphism (suggestion III). I also strongly believe that my concerns about the current limited power of the conclusions presented by the authors can be addressed by adding the suggested analyses. This could be quite a lot of work, but the analyses do not need to be performed in the full parameter space but are mandatory for the paper to be of the highest scientific quality. Here is a short list of the analyses that need to be added (see below for a more detailed description): (I.) In the Gaussian model, add analysis of pathogen peptides evolving by genetic drift by allowing Brownian motion to move the pathogen distribution in allele trait space. (II.) In the bit-string model, add analysis of pathogens evolving by genetic drift. (III.) Add simulations with only the Red Queen process operating (e.g. by simulating haploid hosts co-evolving with pathogens). However, it is necessary that all current assumptions of the bit-string model remain unchanged (including the relationship between infection outcome and host fitness), except for the addition of pathogens co-evolving with hosts.

We appreciate the reviewer’s thoughtful feedback and understand the concerns raised regarding the assumptions and the lack of explicit reference to the Red Queen process. We would like to address these points as follows (see more detailed responses to the specific comments below):

Assumptions used to model heterozygote advantage (HA)

We recognize that some of the assumptions in our model differ from those used in previous studies. However, as we clarify in the detailed responses below, our main focus is on demonstrating that HA can, in fact, maintain high MHC polymorphism under the assumptions of our model. We have further explained that assumptions (a) and (b) together enable HA to maintain MHC diversity, which is in contrast to previous work that has concluded otherwise.

Pathogen co-evolution and the Red Queen process

We agree that pathogen co-evolution and the Red Queen process play an important role in maintaining MHC polymorphism. However, adding simulations that explicitly model the Red Queen process and pathogen evolution via genetic drift would shift the focus of the manuscript and require substantial additional analyses. Instead, we have expanded the Discussion section to qualitatively address how pathogen co-evolution or drift might influence MHC diversity. We believe this is a more appropriate way to handle the reviewer’s concerns within the current scope of the paper.

We hope that our clarifications and expanded discussion address the reviewer’s concerns without detracting from the manuscript's primary focus on heterozygote advantage.

- The authors do not communicate, or perhaps do so in a way that can be easily overlooked by the reader, the fundamental conceptual challenge associated with testing the role of heterozygote advantage in host-pathogen evolution. By this, I mean that heterozygote advantage can only be modelled or tested in the absence of host-pathogen coevolution, because coevolution generates negative frequency-dependent selection (the Red Queen process). Without this explanation, the fact that pathogens do not evolve in the model may seem ridiculous to the reader. This aspect needs to be clear from the point at which the model is described, or perhaps even from the introduction.

We agree with the reviewer and have now expanded the explanation in the Introduction as well as the Model section [L52-54, 105-110]. We clarify both the reasoning behind assuming a constant pathogen community and its implications for ensuring that heterozygote advantage is the primary mechanism responsible for maintaining polymorphism.

- In this model, pathogens do not reproduce, and I have explained above why authors cannot indeed link pathogen fitness to the host immune response if they want to test heterozygote advantage in the absence of the Red Queen process. However, as I understood from the description, pathogens in the bit-string model consist of random pools of peptides: "In the bit-string model, the m pathogens are each given npep randomly drawn bit-strings", and in the Gaussian model, the centres of the pathogens do not change their position in the allele trait space. The consequence of this assumption would be an evolutionary process that stops far in the future (pathogens are constant over time). Even if I am wrong and pathogens are randomly generated in each generation, this will still be incorrect because the model artificially assumes the maximum possible genetic diversity of pathogen peptides, which combined with the way fitness is calculated (similar to a geometric mean) would only favour a large number of alleles under heterozygote advantage. Furthermore, such an unrealistically high genetic diversity of oligopeptides simulates pathogen populations of much larger effective size than in real populations. A proper model for modelling heterozygote advantage would require modelling pathogens evolving by genetic drift. Even if the authors do not agree with my point, they should include the scenario in which pathogens evolve by genetic drift, as this has been assumed in previous models addressing the question of the role of heterozygote advantage and Red Queen on MHC polymorphism. Pathogens evolving by genetic drift are easily implemented in the bit-string model.

We would like to clarify that in our bit-string model, pathogens are indeed fixed over time (the first scenario described by the reviewer). We choose random bit-strings once at the beginning of a simulation.

Then we wonder whether there could be a misunderstanding. Our bit-string model is inspired by the work of Borghans et al. (2004), who assume 50 pathogen species, each with 20 peptides and 10 genotypes per pathogen species. The reviewer raises the concern that our model assumes an unrealistically high genetic diversity for pathogen peptides. However, we would like to clarify that our model actually assumes less diversity than Borghans et al. (2004), as we do not allow for genetic variation within a pathogen species. Instead, we assume the minimum of one genotype per pathogen species. Specifically, one can consider three levels of genetic variation: between-species diversity, within-species diversity, and between-peptide diversity within a species. In contrast to Borghans et al. (2004), our model assumes no within-species diversity, allowing only between-species and between-peptide diversity.

As we argue in response to the reviewer’s next comment, we have biological reasons for why we think that in the Gaussian model, stable pathogen optima are a realistic assumption. The bit-string model was added to our manuscript after the first round of reviews as a robustness check for the Gaussian model. Because of this, we prefer to keep the bit-string model and the Gaussian model as similar as possible. Furthermore, as explained above, drift is not easily added to our model as there is no allelic variation within pathogen species that could fluctuate.

In the Gaussian model, I would suggest introducing Brownian motion of pathogen distributions. It is very unrealistic to assume that pathogens would occupy a stationary point in the allelic trait space over so many generations, rather than moving.

We start our response here by first quoting the last paragraph of the result section for the bit-string model: “We note that the Gaussian and bit-string model suggest slightly different interpretations about the nature of the involved pathogens. In the bit-string model, the same MHC-molecule can in principle detect several pathogens (especially for n_pep}>1). This suggests interpreting different pathogens as strains of the same species or different species that are closely related. For the Gaussian model, a single MHC molecule can generally not detect different pathogens with a high probability. This suggests interpreting different pathogens as clearly distinct species, possibly belonging to different higher taxonomic groups.”

An upshot of our interpretation that different pathogens in the Gaussian model belong to different taxonomic groups is that we consider it likely that the pathogen optima stay approximately constant over the time spans considered here.

- Host fitness is calculated using geometric mean mechanics, which forces the MHC allele (in a hypothetic haploid individual) to be located in the middle of the pathogen distributions located in allele trait space. This is a very strong assumption and differs from other theoretical studies of MHC gene polymorphism (e.g. Borghans et al. 2004, Ejsmond and Radwan 2015). Note, that I do not say that this is an incorrect assumption. Perhaps natural systems differ in the way infections affect host fitness. Anyway, the lack of modelling of other balancing selection forces in the presented work, namely the Red Queen (negative frequency dependence), leaves the reader with a big question mark as to the extent to which the result presented by authors is driven by the specificity of the model or the assumption that immune response and fitness scale in a 'geometric mean'-like manner. Thus, I think the Red Queen process scenario needs to be added, and my suggestion should not be taken as a suggestion of adding an additional analysis beyond the scope of this paper. Each model has its own specificity and it is necessary to know what the levels of polymorphism are when the Red Queen process is simulated. My expectation would be that if fitness is calculated in the way the authors did, the Red Queen process will maintain few alleles. This would be a strong message emerging from this study and the previous literature that heterozygote advantage or the Red Queen process can maintain polymorphism depending on the distribution of fitness effects of infections by different pathogen species.

However, we have indeed expanded the discussion to qualitatively address how pathogen evolution and moving pathogen optima might influence MHC diversity [L473-496]. This approach preserves the main focus while addressing the potential role of pathogen co-evolution.

- The authors should comment on the fact that the MHC gene polymorphism in their model would break down in the presence of several highly virulent pathogen species (narrow distributions in the Gaussian model) and under more realistic assumptions for pathogen classes e.g. helminths, i.e. a large number of antigenic oligopeptides (Figure 5 for v=5).

We suspect that the reviewer has misunderstood the meaning of the virulence parameter *v*.

The parameter *v* is proportional to the *inverse* of the width of the Gaussian functions (cf. Equation 1). Contrary to the reviewer's concern, MHC gene polymorphism is actually favoured in the presence of highly virulent pathogen species (high values of *v*), particularly when several such species are present (high values of *m*). This is illustrated in Figure 4 for the Gaussian model, and in Figure 5 for the bit-string model.

Regarding the issue of a large number of antigenic oligopeptides, our bit-string model does indeed suggest that this can impede the evolution of allelic diversity, at least under lower virulence conditions (v ≤ 7, Figure 5). In such scenarios, evolution can converge to a single optimal MHC allele that is capable of detecting all pathogens, preventing the evolution of MHC polymorphism. This is intuitively clear, as a large number of peptides offers more targets for MHC molecules to recognize. We have now expanded our explanation of this topic in the result section to be even clearer [L 345-351].

- The assumption (a) that pathogens are lethal in the absence of an appropriate immune response does not support the role of heterozygote advantage in maintaining MHC gene polymorphism as suggested by the authors. See the model of Ejsmond et al. 2023 ('Adaptive immune response selects for postponed maturation and increased body size'), where hosts with a poor response to pathogens also die, but high MHC gene polymorphism is maintained in particular by the Red Queen process. However, note also that Ejsmond et al. assumed that infections affect fitness proportionally, which is contrary to assumption (b) in the reviewed paper. I would suggest reducing the importance of the first assumption or discussing alternative models that show that the Red Queen process is able to maintain high MHC gene polymorphism despite assumption (a). I believe that the difference in assumption (b) is fundamental to the conclusions derived from this paper and other models.

We would like to clarify two points in response to this comment.

First, and this is unrelated to the work by Ejsmond et al. (2023), we do not claim that – given assumption (a) is fulfilled – only heterozygote advantage can maintain MHC polymorphism. Instead, we agree that other mechanisms, such as Red Queen processes, can maintain high MHC polymorphism with or without assumption (a). Our main point is that assumptions (a) and (b) together can lead to high MHC diversity being maintained by heterozygote advantage alone, while HA was previously thought to be insufficient for this.

Second, we wish to clarify how in our view the model of Ejsmond et al. (2023) aligns with assumption (a) and (b), although we consider their results to not conflict with our results, as they present a Red Queen model, which is outside the scope of our work.

Contrary to the reviewer's suggestion, the model by Ejsmond et al. in our reading fulfils assumption (b) but not assumption (a). In their model, when infections impact survival, they affect an intermediate property *T*, analogous to our concept of “condition”. Individuals are removed if their condition falls below a threshold, which is adjusted each time step to maintain a constant mortality rate. Fitness is therefore not proportional to *T*, but is modelled through a step-function, meaning that individuals well above the threshold are unaffected by small changes in condition, while those near the threshold face significant mortality risk from small decreases in *T*—consistent with our assumption (b).

However, because *T* represents the average performance against all pathogens (Equation 6 in Ejsmond et al. 2023), a single pathogen can only marginally reduce the condition *T*, so their model does not fulfil assumption (a). In our model, assumption (a) holds because a pathogen to which a host has no immune defence can be lethal, even if the host performs well against all other pathogens

- Show information about the proportion of hosts in the population dying per generation under main simulated scenarios

This information is present in the simulation figures (Figures 3 and 5). The distribution of conditions within the population, divided into homozygotes and heterozygotes, are given by the dark and light blue bars, respectively, and how these conditions map to survival is given by the orange line.

- Consider renaming 'allelic trait space' to epitope or agretope space

We prefer to keep the term ‘allelic trait space’, as allelic values move within this space due to mutations, while pathogens are considered stationary. Therefore, we consider it more appropriate to refer to this as the allelic trait space. However, we appreciate the suggestion and have now incorporated the term 'agretope' in the main text to describe the role of MHC molecules in pathogen detection, ensuring that the immunological context is clear [L 25-28].